# Beyond Demographic Parity: Redefining Equal Treatment

## Abstract

Liberalism-oriented political philosophy reasons that all individuals should be treated equally independently of their protected characteristics. Related work in machine learning has translated the concept of *equal treatment* into terms of *equal outcome* and measured it as *demographic parity* (also called *statistical parity*). Our analysis reveals that the two concepts of equal outcome and equal treatment diverge; therefore, demographic parity does not faithfully represent the notion of *equal treatment*. We propose a new formalization for equal treatment by (i) considering the influence of feature values on predictions, such as computed by Shapley values decomposing predictions across its features, (ii) defining distributions of explanations, and (iii) comparing explanation distributions between populations with different protected characteristics. We show the theoretical properties of our notion of equal treatment and devise a classifier two-sample test based on the AUC of an equal treatment inspector. We study our formalization of equal treatment on synthetic and natural data. We release `explanationspace`, an open-source Python package with methods and tutorials.

## 1 Introduction

In philosophy, long-held discussions about what constitutes a fair or an unfair political system have led to established frameworks of distributive justice (Lamont & Favor, 2017; Kymlicka, 2002). From the *egalitarian* school of thought, the *equal opportunity* concept was argued by Rawls (1958). The concept has been translated into computable metrics with the same name (Hardt et al., 2016). From a machine learning perspective, the technical drawback is that metrics for equal opportunity require label annotations for true positive outcomes, which are not always available after the deployment of a model.

The *liberalism* school of thought[1], put forward by scholars such as Friedman et al. (1990) and Nozick (1974), requires *equal treatment* of individuals regardless of their protected characteristics. This concept has been translated by the machine learning literature (Simons et al., 2021; Heidari et al., 2019; Wachter et al., 2020) into the requirement that machine learning predictions should achieve *equal outcomes* for groups with different protected characteristics. The corresponding measure, *demographic parity* (also called *statistical parity*), compares the different distributions of predicted outcomes of a model $f$ for different social groups, e.g., protected vs dominant. We will show, however, that the metric of demographic parity may indicate fairness, although groups are indeed treated differently.

We leave the normative discussion of which philosophical paradigm should be pursued by policy to the discourse in the social sciences and the broad public. However, our analysis contributes to a foundational understanding of fairness in machine learning. Moreover, we remedy the divergence between the liberalism-oriented philosophical requirement of equal treatment and actual measures used in fair machine learning approaches by proposing a novel method for measuring equal treatment.

---

[1] We use the term *liberalism* to refer to the perspective exemplified by Friedman et al. (1990). This perspective can also be referred to as *neoliberalism* or *libertarianism* (Lamont & Favor, 2017; Friedman, 2022).

Comparing different social groups, we measure how non-protected features of individuals interact with the trained model $f$ as explained by Shapley value attributions (Lundberg & Lee, 2017). If two social groups are treated the same, the distributions of interaction behavior, which we call *explanation distributions*, will not be distinguishable. We introduce a tool, the "Equal Treatment Inspector", that implements this idea. When detecting unequal treatment, it explains the features involved in such an inequality, supporting understanding of the causes of unfairness in the machine learning model. In summary, our contributions are:

1. The definition of explanation distributions as a basis for measuring equal treatment.
2. The definition of a novel method for recognizing and explaining unequal treatment.
3. The study of the formal relationship between equal outcome and equal treatment.
4. A novel Classifier Two Sample Test (C2ST) based on the AUC.
5. A study of synthetic and natural data to demonstrate our method and compare with related work.
6. An open-source Python package `explanationspace` implementing the "Equal Treatment Inspector" that is `scikit-learn` compatible together with documentation and tutorials.

## 2 FOUNDATIONS AND RELATED WORK

This section briefly surveys the philosophical and technical foundations of our contribution as well as related work. We also build on Shapley values, which are generally known in the machine learning community, but for self-containedness we provide their mathematical definition in Appendix A.

### 2.1 BASIC NOTATIONS AND FORMAL DEFINITIONS OF FAIRNESS IN RELATED WORK

In supervised learning, a function $f_\theta : X \to Y$, also called a model, is induced from a set of observations, called the training set, $\mathcal{D} = \{(x_1, y_1), \ldots, (x_n, y_n)\} \sim X \times Y$, where $X = \{X_1, \ldots, X_p\}$ represents predictive features and $Y$ is the target feature. The domain of the target feature is $\text{dom}(Y) = \{0, 1\}$ (binary classification) or $\text{dom}(Y) = \mathbb{R}$ (regression). For binary classification, we assume a probabilistic classifier, and we denote by $f_\theta(x)$ the estimate of the probability $P(Y = 1|X = x)$ over the (unknown) distribution of $X \times Y$. For regression, $f_\theta(x)$ estimates $E[Y|X = x]$. We call the projection of $\mathcal{D}$ on $X$, written $\mathcal{D}_X = \{x_1, \ldots, x_n\} \sim X$, the empirical *input distribution*. The dataset $f_\theta(\mathcal{D}_X) = \{f_\theta(x) \mid x \in \mathcal{D}_X\}$ is called the empirical *prediction distribution*.

We assume a feature modeling protected social groups denoted by $Z$, called *protected feature*, and assume it to be binary valued in the theoretical analysis. $Z$ can either be included in the predictive features $X$ used by a model or not. If not, we assume that it is still available for a test dataset. Even without the protected feature in training data, a model can discriminate against the protected groups by using correlated features as a proxy of the protected one (Pedreschi et al., 2008).

We write $A \perp B$ to denote statistical independence between the two sets of random variables $A$ and $B$, or equivalently, between two multivariate probability distributions. We define two common fairness notions and corresponding fairness metrics that quantify a model's degree of discrimination or unfairness (Mehrabi et al., 2022).

**Definition 2.1.** *(Demographic Parity (DP)).* A model $f_\theta$ achieves demographic parity if $f_\theta(X) \perp Z$.

Thus, demographic parity holds if $\forall z. P(f_\theta(X)|Z = z) = P(f_\theta(X))$. For binary $Z$'s, we can derive an unfairness metric as $d(P(f_\theta(X)|Z = 1), P(f_\theta(X))$, where $d(\cdot)$ is a distance between probability distributions.

**Definition 2.2.** *(Equal Opportunity (EO))* A model $f_\theta$ achieves equal opportunity if $\forall z. P(f_\theta(X)|Y = 1, Z = z) = P(f_\theta(X) = 1|Y = 1)$.

As before, we can measure unfairness for binary $Z$'s as $d(P(f_\theta(X)|Y = 1, Z = 1), P(f_\theta(X) = 1|Y = 1))$. Equal opportunity comes with the problem that labels for correct outcomes are required, but difficult or even impossible to collect Ruggieri et al. (2023).

## 2.2 Philosophical Foundations and Computable Fairness Metrics

Political and moral philosophers from the **egalitarian** school of thought often consider *equal opportunity* to be the key promoter of fairness and social justice, providing qualified individuals with equal chances to succeed regardless of their background or circumstances Rawls (1958; 1991), Dworkin (1981a;b), Arneson (1989), Cohen (1989). In fair ML, Hardt et al. (2016) proposed translating equal opportunity into the inter-group difference of the true positive rates. (Heidari et al., 2019) provided a moral framework to ground such a metric of equal opportunity.

The **liberalism** school of thought argues that individuals should be treated equally independently of outcomes Friedman et al. (1990); Nozick (1974). Equal treatment has also been defined as "equal treatment-as-blindness" or neutrality Sunstein (1992); Miller & Howell (1959). From a technical perspective, the notion of *equal treatment* has often been understood as *equal outcomes* and translated to metrics such as demographic parity or statistical parity (used synonymously). As we will analyze in Section 4.2, equal outcomes imply that two demographic groups experience the same distribution of outcomes, even if the first of the two groups have much better prospects for achieving the predicted outcome. Thus, a model $f$ that achieves equal outcome may have to prefer individuals from one group over those from another group, violating the requirement for equal treatment of all individuals. Our metric for equal treatment remedy this drawback.

In Appendix B.1, we provide an illustrative use-case of when equal treatment is in general desired, but neither equal opportunity nor equal outcomes can model this: scientific paper blind reviews.

## 2.3 Related Work

We briefly review related works below. See Appendix B for an in-depth comparison of our work to existing research.

**Measuring Demographic Parity**. Previous work has relied on the notion of equal outcomes and measured DP on the model predictions using statistics such as Mann–Whitney, Kolmogorov-Smirnov or Wasserstein distance (Raji et al., 2020; Kearns et al., 2018; Cho et al., 2020). Other research lines have aimed to measure DP when the protected attribute is a continuous variable (Jiang et al., 2022). We measure DP by using a classifier two-sample test (see later on) of statistical independence.

**Classifier two-sample test (C2ST)**. We introduce a new classifier two-sample test (C2ST) to measure the independence of sets of random variables. While such a kind of approach has been previously explored by Lopez-Paz & Oquab (2017), the novelty of our proposal is to rely on AUC rather than accuracy, with the advantage of a direct application to the case of non-equal distribution of target labels – more in E.1.

**Explainability for fair supervised learning**. Lundberg (2020) apply Shapley values to statistical parity. While there is a slight overlap with our work, their extended abstract is not comparable to our paper wrt. objectives, breadth, and depth – more in Appendix B.3. Other recent lines of work assume knowledge about causal relationships between random variables, such as Grabowicz et al. (2022). Our work does not rely on causal graphs knowledge but exploits the Shapley values' theoretical properties to obtain fairness model auditing guarantees.

## 3 A Model for Monitoring Equal Treatment

### 3.1 Formalizing Equal Treatment

To establish a criterion for equal treatment, we rely on the notion of explanation distributions.

**Definition 3.1.** *(Explanation Distribution)* An explanation function $\mathcal{S} : \mathcal{F} \times X \to \mathbb{R}^p$ maps a model $f_\theta \in \mathcal{F}$ and an input instance $x \in X$ into a vector of reals $\mathcal{S}(f_\theta, x) \in \mathbb{R}^p$. We extend it by mapping an input distribution $\mathcal{D}_X$ into an (empirical) *explanation distribution* as follows: $\mathcal{S}(f_\theta, \mathcal{D}_X) = \{\mathcal{S}(f_\theta, x) \mid x \in \mathcal{D}_X\} \subseteq \mathbb{R}^p$.

We use Shapley values as an explanation function (cf. Appendix A). In Appendix F, we discuss the usage of LIME. Let us introduce next the new fairness notion of Equal Treatment, which considers the independence of the model's explanations with the membership to a social group.

**Definition 3.2.** *(Equal Treatment (ET))* A model $f_\theta$ achieves ET if $\mathcal{S}(f_\theta, X) \perp Z$.

Such a definition characterizes the philosophical notion of Equal Treatment by encoding the "treatment" performed by the model through the attribution of the importance of its input features. As we will see later in Section 4, Equal Treatment is a stronger notion than Demographic Parity since it not only requires that the distributions of the predictions are similar but that the processes of how predictions are made (i.e., the explanations) are also similar.

### 3.2  EQUAL TREATMENT INSPECTOR

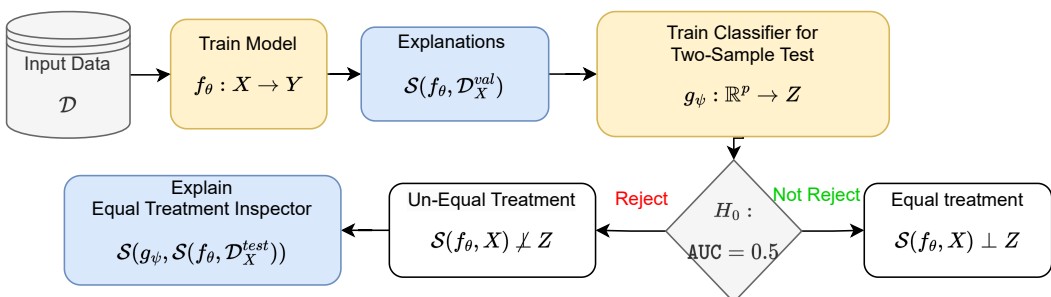

Figure 1: Equal Treatment Inspector workflow. The model $f_\theta$ is learned based on training data, $\mathcal{D}^{tr} = \{(x_i, y_i)\}$, and outputs the explanations $\mathcal{S}(f_\theta, \mathcal{D}_X^{val})$. The C2ST receives the explanations to predict the protected attribute, $Z$ on validation data $\mathcal{D}^{val}$. The AUC of the C2ST classifier $g_\psi$ on test data $\mathcal{D}^{te}$ decides for or against *equal treatment*. We can interpret the driver for unequal treatment on $g_\psi$ with explainable AI techniques.

Our approach is based on the properties of the Shapley values (cf. Appendix A) and on a novel classifier two-sample test. We split the available data into three parts $\mathcal{D}^{tr}, \mathcal{D}^{val}, \mathcal{D}^{te} \subseteq X \times Y$. Here $\mathcal{D}^{tr}$ is the training set of $f_\theta \in \mathcal{F}$ (not required if $f_\theta$ is already trained). Following the intuition above, $\mathcal{D}^{val}$ is used to train another model $g_\psi$, called the "Equal Treatment Inspector". Here, the predictive features are the explanation distribution $\mathcal{S}(f_\theta, \mathcal{D}_{X \setminus Z}^{val})$ (excluding $Z$) and the target feature is the protected feature $Z$. The model $g_\psi$ belongs to a family $\mathcal{G}$, possibly different from $\mathcal{F}$. The parameter $\psi$ optimizes a loss function $\ell$:

$$\psi = \arg\min_{\tilde{\psi}} \sum_{(x,z) \in \mathcal{D}^{val}} \ell(g_{\tilde{\psi}}(\mathcal{S}(f_\theta, x)), z) \tag{1}$$

To evaluate whether there is an equal treatment violation, we perform a statistical test of independence based on the AUC of $g_\psi$ on a test set $\mathcal{D}^{te}$. We also use $\mathcal{D}^{te}$ for testing the approach w.r.t. baselines. Besides detecting fairness violations, a common desideratum is to understand what are the specific features driving such violations. The "Equal Treatment Inspector" $g_\psi$ can provide information on *which features are the cause of the un-equal treatment* either by-design, if it is an interpretable model, or by applying post-hoc explanations techniques, e.g., Shapley values. See Figure 1 for a visualization of the whole workflow.

## 4  THEORETICAL ANALYSIS

Throughout this section, we assume an exact calculation of the Shapley values $\mathcal{S}(f_\theta, x)$ for an instance $x$, possibly for the observational and interventional variants (see (4,5) in Appendix A). In the experimental section, we will use non-IID data and non-linear models.

## 4.1 Equal Treatment Given Shapley Values of Protected Attribute

Can we measure ET by looking only at the Shapley value of the protected feature? The following result considers a linear model (with unknown coefficients) over *independent* features. In such a very simple case, resorting to Shapley values leads to an exact test of both DP and ET, which turn out to coincide. In the following, we write $distinct(\mathcal{D}_X, i)$ for the number of distinct values in the $i$-th feature of dataset $\mathcal{D}_X$, and $\mathcal{S}(f_\beta, \mathcal{D}_X)_i \equiv 0$ if the Shapley values of the $i$-th feature are all 0's.

**Lemma 4.1.** *Consider a linear model $f_\beta(x) = \beta_0 + \sum_j \beta_j \cdot x_j$. Let $Z$ be the $i$-th feature, i.e. $Z = X_i$, and let $\mathcal{D}_X$ be such that $distinct(\mathcal{D}_X, i) > 1$. If the features in $X$ are independent, then $\mathcal{S}(f_\beta, \mathcal{D}_X)_i \equiv 0 \Leftrightarrow f_\beta(X) \perp Z \Leftrightarrow \mathcal{S}(f_\beta, X) \perp Z$.*

*Proof.* It turns out $\mathcal{S}(f_\beta, x)_i = \beta_i \cdot (x_i - E[X_i])$. This holds in general for the interventional variant (5), and assuming independent features, also for the observational variant (4) (Aas et al., 2021). Since $distinct(\mathcal{D}_X, i) > 1$, we have that $\mathcal{S}(f_\beta, \mathcal{D}_X)_i \equiv 0$ iff $\beta_i = 0$. By independence of $X$, this is equivalent to $f_\beta(X) \perp X_i$, i.e., $f_\beta(X) \perp Z$. Moreover, by the propagation of independence, this is also equivalent to $\mathcal{S}(f_\beta, X) \perp Z$. $\qquad\square$

However, the result does not extend to the case of dependent features.

**Example 4.1.** Consider $Z = X_2 = X_1^2$, and the linear model $f_\beta(x_1, x_2) = \beta_0 + \beta_1 \cdot x_1$ with $\beta_1 \neq 0$ and $\beta_2 = 0$, i.e., the protected feature is not used by the model. In the interventional variant, the uninformativeness property implies that $\mathcal{S}(f_\beta, x)_2 = 0$. However, this does not mean that $Z = X_2$ is independent of the output because $f_\beta(X_1, X_2) = \beta_0 + \beta_1 \cdot X_1 \not\perp X_2$. In the observational variant, Aas et al. (2021) show that:

$$val(T) = \sum_{i \in N \setminus T} \beta_i \cdot E[X_i | X_T = x_T^\star] + \sum_{i \in T} \beta_i \cdot x_i^\star$$

from which, we calculate: $\mathcal{S}(f_\beta, x^\star)_2 = \frac{\beta_1}{2} E[X_1 | X_2 = x_2^\star]$. We have $\mathcal{S}(f_\beta, \mathcal{D}_X)_2 \equiv 0$ iff $E[X_1 | x_2 = x_2^\star] = 0$ for all $x^\star$ in $\mathcal{D}_X$. For the marginal distribution $P(X_1 = v) = 1/4$ for $v = 1, -1, 2, -2$, and considering that $X_2 = X_1^2$, it holds that $E[X_1 | x_2 = v] = 0$ for all $v$. Thus $\mathcal{S}(f, \mathcal{D}_X)_2 \equiv 0$. However, again $f_\beta(X_1, X_2) = \beta_0 + \beta_1 \cdot X_1 \not\perp X_2$.

The counterexample shows that focusing only on the Shapley values of the protected feature is not a viable way to prove DP of a model – and, a fortiori, neither to prove ET of the model, as will show in Lemma 4.2.

## 4.2 Equal Treatment vs Equal Outcomes vs Fairness of the Input

We start by observing that *equal treatment* (independence of the explanation distribution from the protected attribute) is a sufficient condition for *equal outcomes* measured as demographic parity (independence of the prediction distribution from the protected attribute).

**Lemma 4.2.** *If $\mathcal{S}(f_\theta, X) \perp Z$ then $f_\theta(X) \perp Z$.*

*Proof.* By the propagation of independence in probability distributions, the premise implies $(\sum_i \mathcal{S}_i(f_\theta, X) + c) \perp Z$ where $c$ is any constant. By setting $c = E[f(X)]$ and by the efficiency property (6), we have the conclusion. $\qquad\square$

Therefore, a DP violation (on the prediction distribution) is also a ET violation (in the explanation distribution). ET accounts for a stricter notion of fairness. The other direction does not hold. We can have dependence of $Z$ from the explanation features, but the sum of such features cancels out resulting in perfect DP on the prediction distribution. This issue is also known as Yule's effect (Ruggieri et al., 2023).

**Example 4.2.** Consider the model $f(x_1, x_2) = x_1 + x_2$. Let $Z \sim Ber(0.5)$, $A \sim U(-3, -1)$, and $B \sim N(2, 1)$ be independent, and let us define:

$$X_1 = A \cdot Z + B \cdot (1 - Z) \quad X_2 = B \cdot Z + A \cdot (1 - Z)$$

We have $f(X_1, X_2) = A + B \perp Z$ since $A, B, Z$ are independent. Let us calculate $\mathcal{S}(f, X)$ in the two cases $Z = 0$ and $Z = 1$. If $Z = 0$, we have $f(X_1, X_2) = B + A$, and then $\mathcal{S}(f, X)_1 = B - E[B] = B - 2 \sim N(0, 1)$ and $\mathcal{S}(f, X)_2 = A - E[A] = A + 2 \sim U(-1, 1)$. Similarly, for $Z = 1$, we have $f(X_1, X_2) = A + B$, and then $\mathcal{S}(f, X)_1 = A - E[A] = A + 2 \sim U(-1, 1)$ and $\mathcal{S}(f, X)_2 = B - E[B] = B - 2 \sim N(0, 1)$. This shows:

$$P(\mathcal{S}(f, X)|Z = 0) \neq P(\mathcal{S}(f, X)|Z = 1)$$

and then $\mathcal{S}(f, X) \not\perp Z$. Notice this example holds both for the interventional and the observational cases, as we exploited Shapley values of a linear model over independent features, namely $A, B, Z$.

Statistical independence between the input $X$ and the protected attribute $Z$, i.e., $X \perp Z$, is another fairness notion. It targets fairness of the (input) datasets, disregarding the model $f_\theta$. For fairness-aware training algorithms, which are able not to (directly or indirectly) rely on $Z$, violation of such a notion of fairness does not imply ET violation nor DP violation.

**Example 4.3.** Let $X = X_1, X_2, X_3$ be independent features such that $E[X_1] = E[X_2] = E[X_3] = 0$, and $X_1, X_2 \perp Z$, and $X_3 \not\perp Z$. The target feature is defined as $Y = X_1 + X_2$, hence it is also independent from $Z$. Assume a linear regression model $f_\beta(x_1, x_2, x_3) = \beta_1 \cdot x_1 + \beta_2 \cdot x_2 + \beta_3 \cdot x_3$ trained from a sample data from $(X, Y)$ with $\beta_1, \beta_2 \approx 1$ and $\beta_3 \approx 0$. Intuitively, this occurs when a number of features are collected to train a classifier without a clear understanding of which of them contributes to the prediction. It turns out that $X \not\perp Z$ but, for $\beta_3 = 0$ (which can be obtained by some fairness regularization method (Kamishima et al., 2011)), we have $f_\beta(X_1, X_2, X_3) = \beta_1 \cdot X_1 + \beta_2 \cdot X_2 \perp Z$. By reasoning as in the proof of Lemma 4.1, we have $\mathcal{S}(f_\beta, X) = (\beta_1 \cdot X_1, \beta_2 \cdot X_2, 0)$ and then $\mathcal{S}(f_\beta, X) \perp Z$. This holds both in the interventional and in the observational variants.

The above represents an example where the input data depends on the protected feature, but the model and the explanations are independent.

### 4.3 Equal Treatment Inspection via Explanation Distributions

#### 4.3.1 Statistical Independence Test via Classifier AUC Test

In this subsection, we introduce a statistical test of independence based on the AUC of a binary classifier. The test of $W \perp Z$ is stated in general form for multivariate random variables $W$ and a binary random variable $Z$ with $dom(Z) = \{0, 1\}$. In the next subsection, we will instantiate it to the case $W = \mathcal{S}(f_\theta, X)$.

Let $\mathcal{D} = \{(w_i, z_i)\}_{i=1}^n$ be a dataset of realizations of the random sample $(W, Z)^n \sim \mathcal{F}^n$ where $\mathcal{F}$ is unknown. The independence $W \perp Z$ can be tested via a two-sample test. In fact, we have $W \perp Z$ iff $P(W|Z) = P(W)$ iff $P(W|Z = 1) = P(W|Z = 0)$. We test whether the positives and negatives instances in $\mathcal{D}$ are drawn from the same distribution by a novel two-sample test, which does not require permutation of data nor equal proportion of positive and negatives as in (Lopez-Paz & Oquab, 2017, Sections 2 and 3). We rely on a probabilistic classifier $f : W \to [0, 1]$, for which $f(w)$ estimates $P(Z = 1|W = w)$, and on its AUC:

$$AUC(f) = E_{(W,Z),(W',Z') \sim \mathcal{F}}[I((Z - Z')(f(W) - f(W')) > 0) + {}^1\!/\!{}_2 \cdot I(f(W) = f(W'))|Z \neq Z'] \tag{2}$$

Under the null hypothesis $H_0 : W \perp Z$, we have $AUC(f) = {}^1\!/\!{}_2$.

**Lemma 4.3.** If $W \perp Z$ then $AUC(f) = {}^1\!/\!{}_2$ for any classifier $f$.

*Proof.* Let us recall the definition of the Bayes Optimal classifier $f_{opt}(w) = P(Z = 1|W = w)$. For any classifier $f$, we have:

$$AUC(f_{opt}) \geq AUC(f) \geq 1 - AUC(f_{opt}) \tag{3}$$

The first bound $AUC(f_{opt}) \geq AUC(f)$ follows because the Bayes Optimal classifier minimizes the Bayes risk (Gao & Zhou, 2015). Assume the second bound does not hold, i.e., for some $f$ we have $AUC(f_{opt}) < 1 - AUC(f)$. Consider the classifier $\bar{f}(w) = 1 - f(w)$.

We have $AUC(\bar{f}) \geq 1 - AUC(f)$, and then $\bar{f}$ would contradict the first bound because $AUC(f_{opt}) < AUC(\bar{f})$.

If $W \perp Z$, then $P(Z = 1|W = w) = P(Z = 1)$, and then $f_{opt}(w)$ is constant. By (2), this implies $AUC(f_{opt}) = 1/2$. By (3), this implies $AUC(f) = 1/2$ for any classifier. $\square$

As a consequence, any statistics to test $AUC(f) = 1/2$ can be used for testing $W \perp Z$. A classical choice is to resort to the Wilcoxon–Mann–Whitney test, which, however, assumes that the distributions of scores for positives and negatives have the same shape. Better alternatives include the Brunner–Munzel test (Neubert & Brunner, 2007) and the Fligner–Policello test (Fligner & Policello, 1981). The former is preferable, as the latter assumes that the distributions are symmetric.

### 4.3.2 Testing for Equal Treatment via an Inspector

We instantiate the previous AUC-based method for testing independence to the case of testing for Equal Treatment via an ET Inspector.

**Theorem 4.4.** *Let $g_\psi : \mathcal{S}(f_\theta, X) \to [0, 1]$ be an "Equal Treatment Inspector" for the model $f_\theta$, and $\alpha$ a significance level. We can test the null hypothesis $H_0 : \mathcal{S}(f_\theta, X) \perp Z$ at $100 \cdot (1 - \alpha)\%$ confidence level using a test statistics of $AUC(g_\psi) = 1/2$.*

*Proof.* Under $H_0$, by Lemma 2 with $W = \mathcal{S}(f_\theta, X)$ and $f = g_\psi$, we have $AUC(g_\psi) = 1/2$. $\square$

Results of such a test can include $p$-values of the adopted test for $AUC(g_\psi) = 1/2$. Alternatively, confidence intervals for $AUC(g_\psi)$ can be reported, as returned by the Brunner–Munzel test or by the methods (DeLong et al., 1988; Cortes & Mohri, 2004; Gonçalves et al., 2014).

### 4.3.3 Explaining Un-Equal Treatment

The following example showcases one of our main contributions: detecting *the sources* of un-equal treatment through interpretable by-design (linear) inspectors. Here, we assume that the model is also linear. In the Appendix E.4, we will experiment with non-linear models.

**Example 4.4.** Let $X = X_1, X_2, X_3$ be independent features such that $E[X_1] = E[X_2] = E[X_3] = 0$, and $X_1, X_2 \perp Z$, and $X_3 \not\perp Z$. Given a random sample of i.i.d. observations from $(X, Y)$, a linear model $f_\beta(x_1, x_2, x_3) = \beta_0 + \beta_1 \cdot x_1 + \beta_2 \cdot x_2 + \beta_3 \cdot x_3$ can be built by OLS (Ordinary Least Square) estimation, possibly with $\beta_1, \beta_2, \beta_3 \neq 0$. By reasoning as in the proof of Lemma 4.1, $\mathcal{S}(f_\beta, x)_i = \beta_i \cdot x_i$. Consider now a linear ET Inspector $g_\psi(s) = \psi_0 + \psi_1 \cdot s_1 + \psi_2 \cdot s_2 + \psi_3 \cdot s_3$, which can be written in terms of the $x$'s as: $g_\psi(x) = \psi_0 + \psi_1 \cdot \beta_1 \cdot x_1 + \psi_2 \cdot \beta_2 \cdot x_2 + \psi_3 \cdot \beta_3 \cdot x_3$. By OLS estimation properties, we have $\psi_1 \approx cov(\beta_1 \cdot X_1, Z)/var(\beta_1 \cdot X_1) = cov(X_1, Z)/(\beta_1 \cdot var(X_1)) = 0$ and analogously $\psi_2 \approx 0$. Finally, $\psi_3 \approx cov(X_3, Z)/(\beta_3 \cdot var(X_3)) \neq 0$. In summary, the coefficients of $g_\psi$ provide information about which feature contributes (and how much it contributes) to the dependence between the explanation $\mathcal{S}(f_\beta, X)$ and the protected feature $Z$. Notice that also $f_\beta(X) \not\perp Z$, but we cannot explain which features contribute to such a dependence by looking at $f_\beta(X)$, since $\beta_i \approx cov(X_i, Y)/var(X_i)$ can be non-zero also for $i = 1, 2$.

## 5 Experimental Evaluation

We perform measures of equal treatment by systematically varying the model $f$, its parameters $\theta$, and the input data distributions $\mathcal{D}_X$. We complement experiments described in this section by adding further experimental results in the Appendix that *(i)* compare the different types of Shapley values estimation (Appendix C), *(ii)* add experiments on further natural datasets (Appendix D), *(iii)* exhibit a larger range of modeling choices (Appendix E.3), *(iv)* compare AUC vs accuracy for the C2ST independence test (Appendix E.1), *(v)* extend the comparison against DP (Appendix E.5) and *(vi)* include LIME as an explanation method (Appendix F).

We adopt xgboost (Chen & Guestrin, 2016) for the model $f_\theta$, and logistic regression for the inspectors. We compare the AUC performances of several inspectors: $g_\psi$ (see Eq. 1) for ET (see Def. 3.2), $g_v$ for DP (see Def. 2.1), $g_\Upsilon$ for fairness of the input (i.e., $X \perp Z$ as discussed

in Section 4.2), and a combination $g_\phi$ of the last two inspectors to test $f_\theta(X), X \perp Z$. These are the formal definitions:

$$\Upsilon = \arg\min_{\tilde{\Upsilon}} \sum_{(x,z)\in\mathcal{D}^{val}} \ell(g_{\tilde{\Upsilon}}(x), z) \quad \upsilon = \arg\min_{\tilde{\upsilon}} \sum_{(x,z)\in\mathcal{D}^{val}} \ell(g_{\tilde{\upsilon}}(f_\theta(x)), z)$$

$$\phi = \arg\min_{\tilde{\phi}} \sum_{(x,z)\in\mathcal{D}^{val}} \ell(g_{\tilde{\phi}}(f_\theta(x), x), z)$$

## 5.1 Experiments with Synthetic Data

We generate synthetic datasets by first drawing $10,000$ samples from normally distributed features $X_1 \sim N(0,1), X_2 \sim N(0,1), (X_3, X_4) \sim N\left(\begin{bmatrix} 0 \\ 0 \end{bmatrix}, \begin{bmatrix} 1 & \gamma \\ \gamma & 1 \end{bmatrix}\right)$. Then, we define a binary protected feature $Z$ with values $Z = 1$ if $X_4 > 0$ and $Z = 0$ otherwise. We compare the methods and baselines while varying the correlation $\gamma = r(X_3, Z)$ from 0 to 1. We define two experimental scenarios below. In both of them, the model $f_\beta$ is a function over the domain of the features $X_1, X_2, X_3$ only.

**Indirect Case:** *Unfairness in the data and in the model.* We consider all of the three features in the dataset $X_1, X_2, X_3$. This gives rise to unfairness of the input parameterized by $\gamma = r(X_3, Z)$. To generate DP violation in the model, we create the target $Y = \sigma(X_1 + X_2 + X_3)$, where $\sigma$ is the logistic function.

**Uninformative Case:** *Unfairness in the data and fairness in the model.* The unfairness in the input data remains the same as in the previous case, while we now remove unfairness in the model. The target feature is now defined as $Y = \sigma(X_1 + X_2)$. The $\gamma$ parameter controls unfairness in the dataset, which should not be captured by the model, since $X_1, X_2 \perp Z$ implies $Y \perp Z$ by propagation of independence.

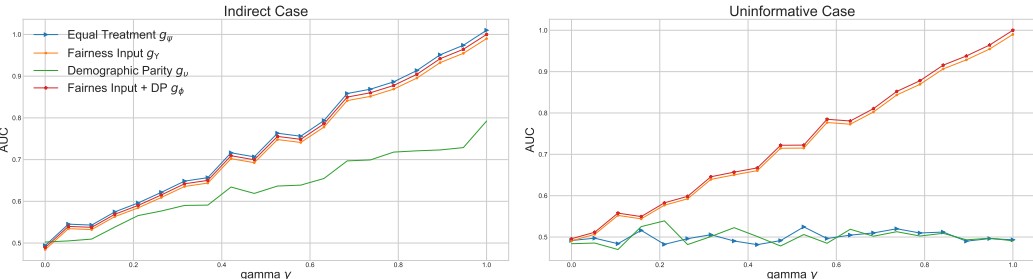

Figure 2: In the "Indirect case" (left): good unfairness detection methods should follow a increasing steady slope to capture the fairness violation; the DT inspector appears less sensitive due to the low dimensionality of its input. In the "Uninformative case" (right): good unfairness detection methods should remain constant with an AUC $\approx 0.5$; the inspectors based on input data ($g_\Upsilon$ and $g_\phi$) flag a false positive case of unfairness.

In Figure 2, we compare the AUC performances of the different inspectors on synthetic data split into $1/3$ for training the model, $1/3$ for training the inspectors and $1/3$ for testing them. Overall, the ET inspector $g_\psi$ is able to detect unfairness in both scenarios. The DP inspector $g_\upsilon$ works fine in the indirect case, but it is not sensitive to unfairness both in the data and in the model in the indirect case. Finally, the inspectors $g_\Upsilon$ and $g_\phi$ detect unfairness in the input but not in the model. Further experiments are shown in Appendix E.4 to investigate the contribution of the explanation distribution features, namely the $\mathcal{S}(f_\theta, x)_i$'s, to the ET inspector $g_\psi$.

## 5.2 Use Case: ACS US Income Data

We experiment here with the ACS Income dataset[2] (Ding et al., 2021), and in the Appendix D with three other ACS datasets. The fairness notions are tested against all pairs of groups

---

[2]ACS PUMS documentation: https://www.census.gov/programs-surveys/acs/microdata/documentation.html

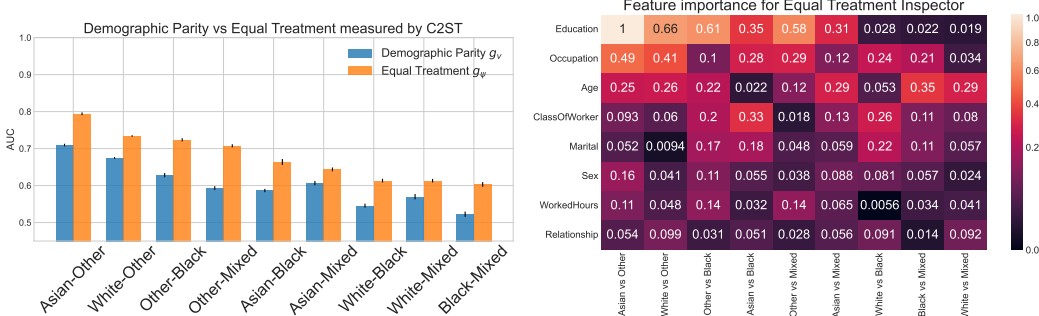

Figure 3: In the left figure, a comparison of ET and DP measures on the US Income data. The AUC range for ET is notably wider, and aligning with the theoretical section, there are indeed instances where DP fails to identify discrimination that ET successfully detects. For a detailed statistical analysis, please refer to Appendix E.5. Right figure provides insight into the influential features contributing to unequal treatment. Higher feature values correspond to a greater likelihood of these features being the underlying causes of unequal treatment.

from the protected attribute "Race". Figure 3 (left) shows the AUC performances of the ET inspector $g_\psi$ and the DT inspector $g_v$. The standard deviation of the AUC is calculated over 30 bootstrap runs, each one splitting the data into ⅓ for training the model, ⅓ for training the inspectors and ⅓ for testing them. In the Appendix E.1, the results of the C2ST test of Section 4.1 are reported. The AUCs for the EP inspectors are greater than for the DP inspectors, as expected due to Lemmma 4.2.

Figure 3 (right) shows the Wasserstein distance between the coefficients of the linear regressor $g_\psi$ compared to a baseline where groups are assigned at random in the input dataset. This feature importance post-hoc explanation method provides insights into the impact of different features as sources of unfairness. We observe "Education" as a highly discriminatory proxy while the role of the feature "Worked Hours Per Week" is less relevant. This allows us to identify areas where adjustments or interventions may be needed to move closer to the ideal of equal treatment.

## 6 Conclusions

We introduced a novel approach for fairness in machine learning by measuring *equal treatment*. While related work reasoned over model predictions to measure *equal outcomes*, our notion of equal treatment is more fine-grained, accounting for the usage of attributes by the model via explanation distributions. Consequently, equal treatment implies equal outcomes, but the converse is not necessarily true, which we confirmed both theoretically and experimentally.

This paper also seeks to improve the understanding of how theoretical concepts of fairness from liberalism-oriented political philosophy align with technical measurements. Rather than merely comparing one social group to another based on disparities within decision distributions, our concept of equal treatment takes into account differences through the explanation distribution of all non-protected attributes, which often act as proxies for protected characteristics. Implications warrant further techno-philosophical discussions. Implications warrant further techno-philosophical discussions.

**Limitations:** Political philosophical notions of distributive justice are more complex than we can account for in this paper. Our research has focused on tabular data using Shapley values, which allow for theoretical guarantees but may differ from their computational approximations. It is possible that alternative AI explanation techniques, such as feature attribution methods, logical reasoning, argumentation, or counterfactual explanations, could be useful and offer their unique advantages to definitions of equal treatment. It is important to note that employing fair AI techniques does not necessarily ensure fairness in socio-technical systems based on AI, as stated in Kulynych et al. (2020).

REPRODUCIBILITY STATEMENT

To ensure the reproducibility of our results, we make publicly available at https://anonymous.4open.science/r/xAIAuditing-F6F9/README.md: the data, the data preparation routines, the source code, and the code of experimental results. Also, the open-source Python package `explanationspace` https://anonymous.4open.science/r/explanationspace-B4B1/README.md will be released. We use default `scikit-learn` parameters (Pedregosa et al., 2011), unless stated otherwise. Our experiments were run on a 4 vCPU server with 32 GB RAM.

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

CONTENTS

## A    DEFINITION AND PROPERTIES OF SHAPLEY VALUES

Explainability has become an important concept in legal and ethical guidelines for data processing and machine learning applications (Selbst & Barocas, 2018). A wide variety of methods have been developed, aiming to account for the decision of algorithmic systems (Guidotti et al., 2019; Mittelstadt et al., 2019; Arrieta et al., 2020). One of the most popular approaches to explainability in machine learning is Shapley values.

Shapley values are used to attribute relevance to features according to how the model relies on them (Lundberg et al., 2020; Lundberg & Lee, 2017; Rozemberczki et al., 2022). Shapley values are a coalition game theory concept that aims to allocate the surplus generated by the grand coalition in a game to each of its players (Shapley, 1997).

For set of players $N = \{1, \ldots, p\}$, and a value function $\text{val} : 2^N \to \mathbb{R}$, the Shapley value $\mathcal{S}_j$ of the $j$'th player is defined as the average marginal contribution of player $j$ in all possibles coalitions of players:

$$\mathcal{S}_j = \sum_{T \subseteq N \setminus \{j\}} \frac{|T|!(p - |T| - 1)!}{p!} (\text{val}(T \cup \{j\}) - \text{val}(T))$$

In the context of machine learning models, players correspond to features $X_1, \ldots, X_p$, and the contribution of the feature $X_j$ is with reference to the prediction of a model $f$ for an instance $x^\star$ to be explained. Thus, we write $\mathcal{S}(f, x^\star)_j$ for the Shapley value of feature $X_j$ in the prediction $f(x^\star)$. We denote by $\mathcal{S}(f, x^\star)$ the vector of Shapely values $(\mathcal{S}(f, x^\star)_1, \ldots, \mathcal{S}(f, x^\star)_p)$.

There are two variants for the term $\text{val}(T)$ (Aas et al., 2021; Chen et al., 2020; Zern et al., 2023): the *observational* and the *interventional*. When using the observational conditional expectation, we consider the expected value of $f$ over the joint distribution of all features conditioned to fix features in $T$ to the values they have in $x^\star$:

$$\text{val}(T) = E[f(x^\star_T, X_{N \setminus T}) | X_T = x^\star_T] \tag{4}$$

where $f(x^\star_T, X_{N \setminus T})$ denotes that features in $T$ are fixed to their values in $x^\star$, and features not in $T$ are random variables over the joint distribution of features. Opposed, the interventional conditional expectation considers the expected value of $f$ over the marginal distribution of features not in $T$:

$$\text{val}(T) = E[f(x^\star_T, X_{N \setminus T})] \tag{5}$$

In the interventional variant, the marginal distribution is unaffected by the knowledge that $X_T = x^\star_T$. In general, the estimation of (4) is difficult, and some implementations (e.g., SHAP) actually consider (5) as the default one. In the case of decision tree models, TreeSHAP offers both possibilities.

The Shapley value framework is the only feature attribution method that satisfies the properties of efficiency, symmetry, uninformativeness and additivity (Molnar, 2019; Shapley, 1997; Winter, 2002; Aumann & Dreze, 1974). We recall next the key properties of efficiency and uninformativeness:

**Efficiency.** Feature contributions add up to the difference of prediction for $x^\star$ and the expected value of $f$:

$$\sum_{j \in N} \mathcal{S}(f, x^\star)_j = f(x^\star) - E[f(X)]) \tag{6}$$

The following property only holds for the interventional variant (e.g., for SHAP values), but not for the observational variant.

**Uninformativeness.** A feature $X_j$ that does not change the predicted value (i.e., for all $x, x'_j$: $f(x_{N \setminus \{j\}}, x_j) = f(x_{N \setminus \{j\}}, x'_j)$) has a Shapley value of zero, i.e., $\mathcal{S}(f, x^\star)_j = 0$.

In the case of a linear model $f_\beta(x) = \beta_0 + \sum_j \beta_j \cdot x_j$, the SHAP values turns out to be $\mathcal{S}(f, x^\star)_i = \beta_i(x_i^\star - \mu_i)$ where $\mu_i = E[X_i]$. For the observational case, this holds only if the features are independent (Aas et al., 2021).

## B  DETAILED RELATED WORK

This section provides an in-depth review of the related theoretical work that informs our research. We contextualize our contribution within the broader field of explainable AI and fairness auditing. We discuss the use of fairness measures such as demographic parity, as well as explainability techniques like Shapley values and counterfactual explanations.

### B.1  FAIRNESS NOTIONS: PAPER BLIND REVIEWS USE CASE

To illustrate the difference between equal opportunity, equal outcomes, and equal treatment, based on the previously discussed framework, we consider the example of conference papers' blind reviews and focus on the protected attribute of the country of origin of the paper's author, comparing Germany and the United Kingdom.

For *equal opportunity*, we quantify fairness by the true positive rate (cf. Definition 2.2). In words, it is the acceptance ratio given that the quality of the paper is high. Achieving equal opportunity will imply that these ratios are similar between the two countries. In blind reviews, the purpose is to evaluate the paper's quality and the research's merit without being influenced by factors such as the author's identity, affiliations, background or country. If we were to enforce equal opportunity in this use case, we would aim for similar true positive rates for submissions from different countries. However, this approach could lead to unintended consequences, such as unintentionally favouring, reverse discrimination, overcorrection or quotas of affirmative action towards certain countries.

For *equal outcomes*, we require that the distribution of acceptance rates is similar, independently of the quality of the paper (cf. Definition 2.1). Note that the outcomes can have similar rates due to random chance, even if there is a country bias in the acceptance procedure.

For *equal treatment*, we require that the contributions of the features used to make a decision on paper's acceptance has similar distributions (cf. Definition 3.2). Equality of treatment through blindness is more desirable than equal opportunity or equal outcomes because it ensures that all submissions are evaluated solely on the basis of their quality, without any bias or discrimination towards any particular country. By achieving equality of treatment through blindness, we can promote fairness and objectivity in the review process and ensure that all papers have an equal chance to be evaluated on their merits.

In comparing our introduced measure of *equal treatment* with *equal outcomes* (or demographic or statistical parity, used as synonymous), we note that the latter looks at the distributions of predictions and measures their similarity. Equal treatment goes a step further by evaluating whether the contribution of features to the decision, is similar. Our definition of *equal treatment* implies the notion of *equal outcome*, but the converse is not necessarily true, as we showed in Section 4.3.1.

## B.2 Measuring Fairness

Selecting a measure to compare fairness between two sensitive groups has been a highly discussed topic, where results such as (Chouldechova, 2017; Hardt et al., 2016; Kleinberg et al., 2017), have highlighted the impossibility to satisfy simultaneously three type of fairness measures: demographic parity (Dwork et al., 2012), equalized odds (Hardt et al., 2016), and predictive parity (Corbett-Davies et al., 2017; Ruf & Detyniecki, 2021; Wachter et al., 2020).

Previous work has relied on measuring and calculating demographic parity on the model predictions (Raji et al., 2020; Kearns et al., 2018), or on the input data (Fabbrizzi et al., 2022; Yang et al., 2020; Zhao et al., 2017). In this work, we perform equal treatment measures on the explanation distribution, which measures that each feature contributes equally to the prediction, which differs from the previous notions.

In this work, we focus on Equal Treatment (ET), as this fairness metric does not require a ground truth target variable, allowing for our method to work in its absence (Aka et al., 2021), and under distribution shift conditions (Mougan et al., 2022) where model performance metrics are not feasible to calculate (Garg et al., 2021; 2022; Mougan & Nielsen, 2023). Demographic Parity requires independence of the model's output from the protected features, written $f_\theta(X) \perp Z$, while Equal Treatment requires independence accross the feature attributions of the model $\mathcal{S}(f_\theta(X), X) \perp Z$.

## B.3 Explainability and fair supervised learning

The intersection of fairness and explainable AI has been an active topic in recent years. The work most close to our approach is Lundberg (2020) where Shapley values are aimed at testing for demographic parity. This concise workshop paper emphasizes the importance of "decomposing a fairness metric among each of a model's inputs to reveal which input features may be driving any observed fairness disparities". In terms of statistical independence, the approach can be rephrased as decomposing $f_\theta(X) \perp Z$ by examining $S(f_\theta, X)_i \perp Z$ for $i \in [1, p]$. Actually, the paper limits to consider difference in means, namely testing for $E[\mathcal{S}(f_\theta, X)_i | Z = 1] \neq E[\mathcal{S}(f_\theta, X)_i | Z = 0]$. Our approach goes beyond this, as we consider different distributions, and introduce the ET fairness notion for that. On the contrary, Lundberg (2020) claims a decomposition method specific of DP. However, the decomposition method proposed is not sufficient nor necessary to prove DP, as showed next.

**Lemma B.1.** $f_\theta(X) \perp Z$ is neither implied by nor it implies $(\mathcal{S}(f_\theta, X)_i \perp Z$ for $i \in [1, p])$.

*Proof.* Consider $f_\theta(X_1, X_2) = X_1 - X_2$ with $X_1, X_2 \sim \texttt{Ber}(0.5)$ and $Z = 1$ if $X_1 = X_2$, and $Z = 0$ otherwise. Hence $Z \sim \texttt{Ber}(0.5)$. We have $\mathcal{S}(f_\theta, X_1) = X_1 \perp Z$ and $\mathcal{S}(f, X_2) = -X_2 \perp Z$. However, $f_\theta(X_1, X_2) = X_1 - X_2$ does not satisfy $f_\theta(X_1, X_2) \perp Z$, e.g., $P(Z = 0 | f_\theta(X_1, X_2) = 0) = P(Z = 0 | X_1 - X_2 = 0) = 1$. Example 4.2 illustrates a case where $f_\theta(X) \perp Z$ yet $\mathcal{S}(f_\theta, X)_1$ and $\mathcal{S}(f_\theta, X)_2$ are not independent of $Z$. ☐

Our approach to ET considers the independence of the *multivariate* distribution of $\mathcal{S}(f, X)$ with respect to $Z$, rather than the independence of each marginal distribution $\mathcal{S}(f_\theta, X)_i \perp Z$. With such a definition, we obtain a sufficient condition for DP, as shown in Lemma 4.2.

Stevens et al. (2020) presents an approach based on adapting the Shapley value function to explain model unfairness. They also introduce a new meta-algorithm that considers the problem of learning an additive perturbation to an existing model in order to impose fairness. In our work, we do not adopt the Shapley value function. Instead, we use the theoretical Shapley properties to provide fairness auditing guarantees. Our "Equal Treatment Inspector" is not perturbation-based but uses Shapley values to project the model to the explanation distribution, and then measures *un-equal treatment*. It also allows us to pinpoint what are the specific features driving this violation.

Grabowicz et al. (2022) present a post-processing method based on Shapley values aiming to detect and nullify the influence of a protected attribute on the output of the system. For this, they assume there are direct causal links from the data to the protected attribute and

that there are no measured confounders. Our work does not use causal graphs but exploits the theoretical properties of the Shapley values to obtain fairness model auditing guarantees.

A few works have researched fairness using other explainability techniques such as counterfactual explanations (Kusner et al., 2017; Manerba & Guidotti, 2021; Mutlu et al., 2022). We don't focus on counterfactual explanations but on feature attribution methods that allow us to measure unequal feature contribution to the prediction. Further work can be envisioned by applying explainable AI techniques to the "Equal Treatment Inspector" or constructing the explanation distribution out of other techniques.

### B.4 Classifier Two-Sample Test (C2ST)

The use of classifiers as a statistical tests of independence $W \perp Z$ for a binary $Z$ has been previously explored in the literature (Lopez-Paz & Oquab, 2017). The approach relies on testing accuracy of a classifier trained to distinguish $Z = 1$ (positives) from $Z = 0$ (negatives) given $W = w$. In the null hypothesis that the distributions of positives and negatives are the same, no classifier is better than a random answer with accuracy $1/2$. This assumes equal proportion of instances of the two distributions in the training and test set. Our approach builds on this idea, but it considers testing the AUC instead of the accuracy. Thus, we remove the assumption of equal proportions[3]. We also show in Section E.1 that using AUC may achieve a better power than using accuracy.

Liu et al. (2020) propose a kernel-based approach to two-sample tests classification. Alike work has also been used in Kaggle competitions under the name of "Adversarial Validation" (Ellis, 2023; Guschin et al., 2018), a technique which aims to detect which features are distinct between train and leaderboard datasets to avoid possible leaderboard shakes.

Edwards & Storkey (2016) focuses on removing statistical parity from images by using an adversary that tries to predict the relevant sensitive variable from the model representation and censoring the learning of the representation of the model and data on images and neural networks. While methods for images or text data are often developed specifically for neural networks and cannot be directly applied to traditional machine learning techniques, we focus on tabular data where techniques such as gradient boosting decision trees achieve state-of-the-art model performance (Grinsztajn et al., 2022; Elsayed et al., 2021; Borisov et al., 2021). Furthermore, our model and data projection into the explanation distributions leverages Shapley value theory to provide fairness auditing guarantees. In this sense, our work can be viewed as an extension of their work, both in theoretical and practical applications.

## C    True to the Model or True to the Data?

Many works discuss the application of Shapley values for feature attribution in ML models (Strumbelj & Kononenko, 2014; Lundberg et al., 2020; Lundberg & Lee, 2017; Lundberg et al., 2018). However, the correct way to connect a model to a coalitional game, which is the central concept of Shapley values, is a source of controversy, with two main approaches: an interventional (Aas et al., 2021; Frye et al., 2020; Zern et al., 2023), and an observational formulation of the conditional expectation, see (4,5) (Sundararajan & Najmi, 2020; Datta et al., 2016; Mase et al., 2019).

In the following experiment, we compare the impact of the two approaches on our "Equal Treatment Inspector". We benchmark this experiment on the four prediction tasks based on the US census data (Ding et al., 2021) and use linear models for both the $f_\theta(X)$ and $g_\psi(\mathcal{S}(f_\theta, X))$. We calculate the two variants of Shapley values using the SHAP linear explainer.[4] The comparison will be parametric to a feature perturbation hyperparameter. The interventional SHAP values break the dependence structure between features in the model to uncover how the model would behave if the inputs are changed (as it was an intervention). This option is said to stay "true to the model" meaning it will only give

---

[3]For unequal proportions, one can consider the accuracy of the majority class, but this still make the requirement to know the true proportion of positives and negatives.

[4]https://shap.readthedocs.io/en/latest/generated/shap.explainers.Linear.html

allocation credit to the features that are actually used by the model. On the other hand, the full conditional approximation of the SHAP values respects the correlations of the input features. If the model depends on one input that is correlated with another input, then both get some credit for the model's behaviour. This option is said to say "true to the data", meaning that it only considers how the model would behave when respecting the correlations in the input data (Chen et al., 2020). We will measure the difference between the two approaches by looking at the AUC and at the linear coefficients of the inspector $g_\psi$, for this case only for the pair White-Other. In Table 1 and Table 2, we can see that differences in AUC and coefficients are negligible.

Table 1: AUC comparison of the "Equal Treatment Inspector" between estimating the Shapley values between the interventional and the observational approaches for the four prediction tasks based on the US census dataset. The % column is the relative difference.

|  | Interventional | Correlation | % |
|---|---|---|---|
| Income | 0.736438 | 0.736439 | 1.1e-06 |
| Employment | 0.747923 | 0.747923 | 4.44e-07 |
| Mobility | 0.690734 | 0.690735 | 8.2e-07 |
| Travel Time | 0.790512 | 0.790512 | 3.0e-07 |

Table 2: Linear regression coefficients comparison of the "Equal Treatment Inspector" between estimating the Shapley values between the interventional and the observational approaches for the ACS Income prediction task. The % column is the relative difference.

|  | Interventional | Correlation | % |
|---|---|---|---|
| Marital | 0.348170 | 0.348190 | 2.0e-05 |
| Worked Hours | 0.103258 | -0.103254 | 3.5e-06 |
| Class of worker | 0.579126 | 0.579119 | 6.6e-06 |
| Sex | 0.003494 | 0.003497 | 3.4e-06 |
| Occupation | 0.195736 | 0.195744 | 8.2e-06 |
| Age | -0.018958 | -0.018954 | 4.2e-06 |
| Education | -0.006840 | -0.006840 | 5.9e-07 |
| Relationship | 0.034209 | 0.034212 | 2.5e-06 |

## D    EXPERIMENTS ON DATASETS DERIVED FROM THE US CENSUS

In the main body of the paper, we considered the ACS Income dataset. Here, we experiment with additional datasets derived from the US census database (Ding et al., 2021): ACS Travel Time, ACS Employment and ACS Mobility. We compare fairness of the prediction tasks for pairs of protected attribute groups over the California 2014 district data.

We follow the same methodology as in the experimental Section 5.2. The choice of xgboost (Chen & Guestrin, 2016) for the model $f_\beta$ is motivated as it achieves state-of-the-art performance Grinsztajn et al. (2022); Elsayed et al. (2021); Borisov et al. (2021). The choice of logistic regression for the inspector $g_\psi$ is motivated by its direct interpretability.

### D.1    ACS EMPLOYMENT

The goal of this task is to predict whether an individual, is employed. Figure 4 shows a low DP violation, compared to the other prediction tasks based on the US census dataset. The AUC of the "Equal Treatment Inspector" is ranging from 0.55 to 0.70. For Asian vs Black un-equal treatment we see that there significant variation of the AUC, indicating that the method achieves different values on the bootstrapping folds. Looking at the features driving the ET violation, we see particularly high values when comparing "Asian" and "Black" populations, and for features "Citizenship" and "Employment". On average, the most important features across all group comparisons are also "Education" and "Area". Interestingly, features such as "difficulties on the hearing or seeing", do not play a role.

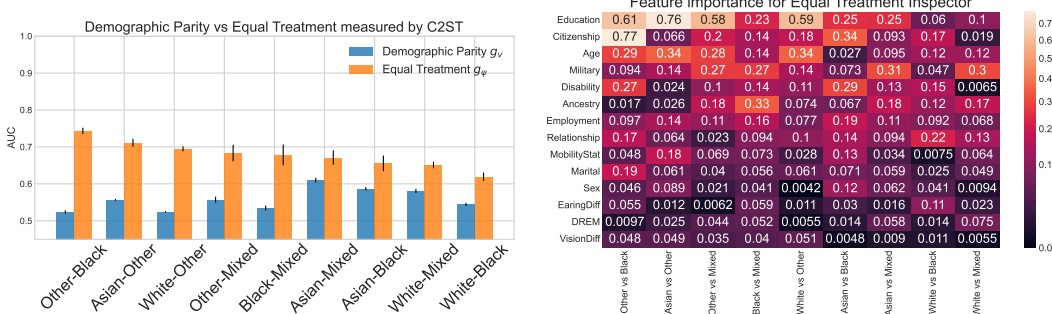

Figure 4: Left: AUC of the inspector for ET and DP, over the district of California 2014 for the ACS Employment dataset. Right: contribution of features to the ET inspector performance.

## D.2 ACS Travel Time

The goal of this task is to predict whether an individual has a commute to work that is longer than 20 minutes. The threshold of 20 minutes was chosen as it is the US-wide median travel time to work based on 2018 data. Figure 5 shows an AUC for the ET inspector in the range of 0.50 to 0.60. By looking at the features, they highlight different ET drivers depending on the pair-wise comparison made. In general, the feature "Education", "Citizenship" and "Area" are the those with the highest difference. Even though for Asian-Black pairwise comparison "Employment" is also one of the most relevant features.

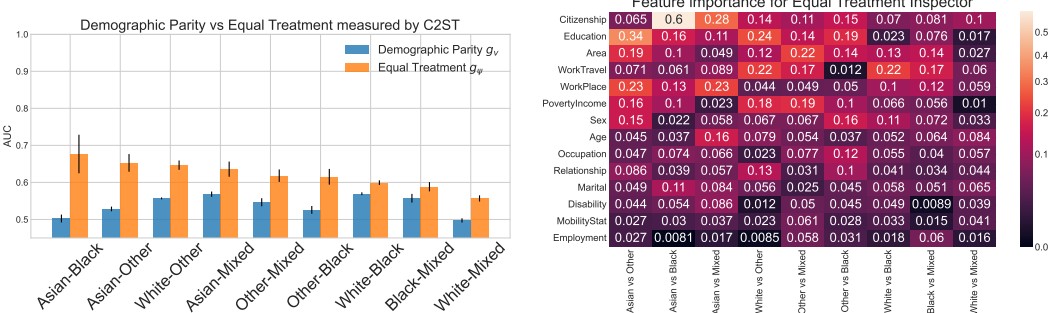

Figure 5: Left: AUC of the inspector for ET and DP, over the district of California 2014 for the ACS Travel Time dataset. Right: contribution of features to the ET inspector performance.

## D.3 ACS Mobility

The goal of this task is to predict whether an individual had the same residential address one year ago, only including individuals between the ages of 18 and 35. This filtering increases the difficulty of the prediction task, as the base rate of staying at the same address is above 90% for the general population (Ding et al., 2021). Figure 6 show an AUC of the ET inspector in the range of 0.55 to 0.80. By looking at the features, they highlight different source of the ET violation depending on the group pair-wise comparison. In general the feature "Ancestry", i.e. "ancestors' lives with details like where they lived, who they lived with, and what they did for a living", plays a high relevance when predicting ET violation.

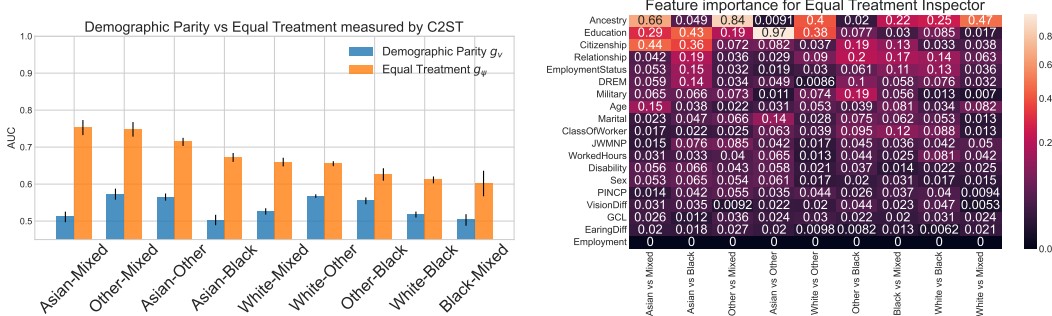

Figure 6: Left: AUC of the inspector for ET and DP, over the district of California 2014 for the ACS Mobility dataset. Right: contribution of features to the ET inspector performance.

## E  ADDITIONAL EXPERIMENTS

In this section, we run additional experiments regarding C2ST, hyperparameters, and models for estimators $f_\theta$ and inspectors $g_\psi$.

### E.1  STATISTICAL INDEPENDENCE TEST VIA CLASSIFIER AUC TEST

We complement the experiments of Section 5.2 by reporting in Table 3 the results of the C2ST for group pair-wise comparisons. As discussed in Section 4.3.1, we perform the statistical test $H_0 : AUC = 1/2$ of the "Equal Treatment Inspector" using a Brunner-Munzel one tailed test against $H_1 : AUC > 1/2$ as implemented in Virtanen et al. (2020). Table 3 reports the empirical AUC on test set, the confidence intervals at 95% confidence level (columns "Low" and "High"), and the p-value of the test. The "Random" row regards a randomly assigned group and represents a baseline for comparison. The statistical tests clearly show that the AUC is significantly different from $1/2$, also when correcting for multiple comparison tests.

Table 3: Results of the C2ST on the "Equal Treatment Inspector".

| Pair | AUC | Low | High | pvalue | Test Statistic |
|------|-----|-----|------|--------|----------------|
| Random | 0.501 | 0.494 | 0.507 | 0.813 | 0.236 |
| White-Other | 0.735 | 0.731 | 0.739 | < 2.2e-16 | 97.342 |
| White-Black | 0.62 | 0.612 | 0.627 | < 2.2e-16 | 27.581 |
| White-Mixed | 0.615 | 0.607 | 0.624 | < 2.2e-16 | 23.978 |
| Asian-Other | 0.795 | 0.79 | 0.8 | < 2.2e-16 | 107.784 |
| Asian-Black | 0.667 | 0.659 | 0.676 | < 2.2e-16 | 38.848 |
| Asian-Mixed | 0.644 | 0.634 | 0.653 | < 2.2e-16 | 28.235 |
| Other-Black | 0.717 | 0.708 | 0.725 | < 2.2e-16 | 48.967 |
| Other-Mixed | 0.697 | 0.688 | 0.707 | < 2.2e-16 | 39.925 |
| Black-Mixed | 0.598 | 0.586 | 0.61 | < 2.2e-16 | 15.451 |

We also compare the power of the C2ST based on the AUC against the two-sample test of Lopez-Paz & Oquab (2017), which is based on accuracy. We generate synthetic datasets where $Y \sim Ber(0.5)$ and $X = (X_1, X_2)$ with positives distributed as $N((\mu, \mu), \Sigma)$ and negatives distributed as $N((-\mu, -\mu), \Sigma)$, where $\Sigma = \begin{bmatrix} 1 & 0.5 \\ 0.5 & 1 \end{bmatrix}$. Thus, the large the $\mu$, the easier is to distinguish the two distributions. Figure 7 reports the power of the AUC-based test vs the accuracy-based test using a logistic regression classifier, estimated by 1000 runs for each of the $\mu$'s ranging from 0.005 to 0.1. The figure highlights that, under such a setting, testing the AUC rather than the accuracy leads to a better power (probability of rejecting $H_0$ when it does not hold).

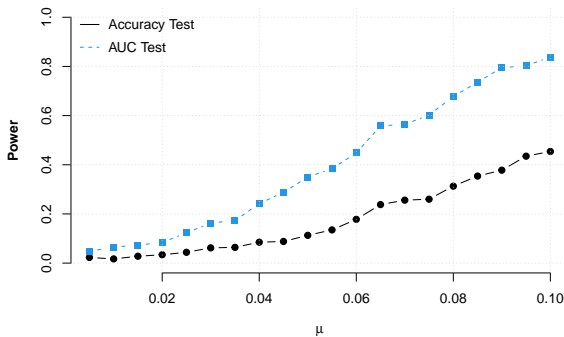

Figure 7: Comparing the power of C2ST based on Accuracy vs AUC.

## E.2 Hyperparameters Evaluation

This section presents an extension to our experimental setup, where we increase the model complexity by varying the model hyperparameters. We use the US Income dataset for the population of the CA14 district. We consider three models for $f_\theta$: Decision Trees, Gradient Boosting, and Random Forest. For the Decision Tree models, we vary the depth of the tree, while for the Gradient Boosting and Random Forest models, we vary the number of estimators. Shapley values are calculated by means of the TreeExplainer algorithm (Lundberg et al., 2020). For the ET inspector $g_\psi$, we consider logistic regession, and XGB.

Figure 8 shows that less complex models, such as Decision Trees with maximum depth 1 or 2, are also less unfair. However, as we increase the model complexity, the unequal treatment of the model becomes more pronounced, achieving a plateau when the model has enough complexity. Furthermore, when we compare the results for different ET inspectors, we observe minimal differences (note that the y-axis takes different ranges).

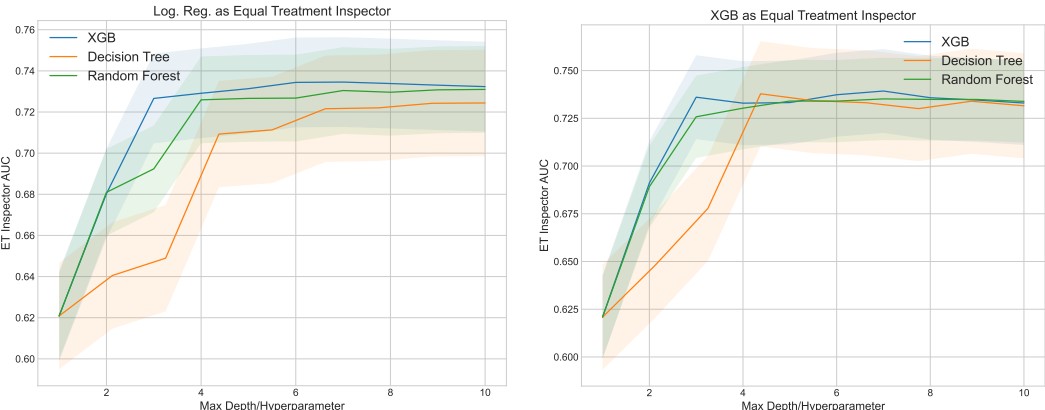

Figure 8: AUC of the inspector for ET, over the district of CA14 for the US Income dataset.

## E.3 Varying Estimator and Inspector

We vary here the model $f_\theta$ and the inspector $g_\psi$ over a wide range of well-known classification algorithms. Table 4 shows that the choice of model and inspector impacts on the measure of Equal Treatment, namely the AUC of the inspector. By Theorem 4.4, the larger the AUC of any inspector the smaller is the $p$-value of the null hypothesis $\mathcal{S}(f_\theta, X) \perp Z$. Therefore, inspectors able to achive the best AUC should be considered. Weak inspectors have lower probability of rejecting the null hypothesis when it does not hold.

| Inspector $g_\psi$ | Model $f_\theta$ | | | | |
|---|---|---|---|---|---|
| | **DecisionTree** | **SVC** | **Logistic Reg.** | **RF** | **XGB** |
| **DecisionTree** | 0.631 | 0.644 | 0.644 | 0.664 | 0.634 |
| **KNN** | 0.737 | 0.754 | 0.75 | 0.744 | 0.751 |
| **Logistic Reg.** | 0.767 | 0.812 | 0.812 | 0.812 | 0.821 |
| **MLP** | 0.786 | 0.795 | 0.795 | 0.813 | 0.804 |
| **RF** | 0.776 | 0.782 | 0.781 | 0.758 | 0.795 |
| **SVC** | 0.743 | 0.807 | 0.807 | 0.790 | 0.810 |
| **XGB** | 0.775 | 0.780 | 0.780 | 0.789 | 0.790 |

Table 4: AUC of the ET inspector for different combinations of models and inspectors.

### E.4 EXPLAINING ET UNFAIRNESS

We complement the results of the experimental Section 5.1 with a further experiment relating the correlation hyperparameter $\gamma$ to the coefficients of an explainable ET inspector. We consider a synthetic dataset with one more feature, by drawing $10,000$ samples from a $X_1 \sim N(0,1)$, $X_2 \sim N(0,1)$, and $(X_3, X_5)$ and $(X_4, X_5)$ following bivariate normal distributions $N\left([0\ 0], [1\ \ \gamma\gamma\ \ 1]\right)$ and $N\left([0\ 0], [1\ \ \gamma0.5\gamma0.5\ \ 1]\right)$, respectively. We define the binary protected feature $Z$ with values $Z = 1$ if $X_5 > 0$ and $Z = 0$ otherwise. As in Section 5.1, we consider two experimental scenarios. In the first scenario, the indirect case, we have unfairness in the data and in the model. The targe feature is $Y = \sigma(X_1 + X_2 + X_3 + X_4)$, where $\sigma$ is the logistic function. In the second scenario, the uninformative case, we have unfairness in the data and fairness in the model. The target feature is $Y = \sigma(X_1 + X_2)$.

Figure 9 shows howt the coefficients of the inspector $g_\psi$ vary with correlation $\gamma$ in both scenario. In the indirect case, coefficients for $\mathcal{S}(f_\theta, X_1)_1$ and $\mathcal{S}(f_\theta, X_1)_2$ correctly attributes zero importance to such variables, while coefficients for $\mathcal{S}(f_\theta, X_1)_3$ and $\mathcal{S}(f_\theta, X_1)_4$ grow linearly with $\gamma$, and with the one for $\mathcal{S}(f_\theta, X_1)_3$ with higher slope as expected. In the uninformative case, coefficients are correctly zero for all variables.

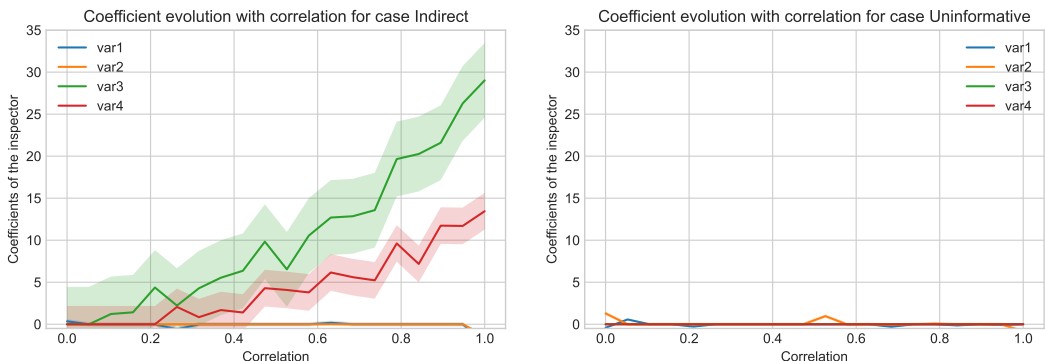

Figure 9: Coefficient of $g_\psi$ over $\gamma$ for synthetic datasets in two experimental scenarios.

### E.5 STATISTICAL COMPARISON OF DEMOGRAPHIC PARITY VERSUS EQUAL TREATMENT

So far, we measured ET and DP fairness usingthe AUC of an inspector, $g_\psi$ and $g_v$ respectively (see Section 5). For DP, however, other probability density distance metrics can be considered, including the p-value of the Kolmogorov–Smirnov (KS) test and the Wasserstein distance. Table 5 reports all such distances in the format "mean $\pm$ stdev" calculated over 100 random sampled datasets. The pairs of group comparisons are sorted by descending AUC values. We highlight in red values below the threshold of 0.05 for the KS test, of 0.55 for the AUC of the C2ST, and of 0.05 for the Wasserstein distance. They represent cases where ET violation occurs, but no DP violation is measured (with different metrics).

Table 5: Comparison of ET and DP measured in differnt ways. Case of ET violaion but no DP violation are highlighted in red.

| Pair | Data | Equal treatment | Demographic Parity | | |
|------|------|------|------|------|------|
| | | C2ST(AUC) | C2ST(AUC) | KS(pvalue) | Wasserstein |
| Asian-Other | Income | $0.794 \pm 0.004$ | $0.709 \pm 0.004$ | $0.338 \pm 0.007$ | $0.256 \pm 0.004$ |
| White-Other | Income | $0.734 \pm 0.002$ | $0.675 \pm 0.003$ | $0.282 \pm 0.003$ | $0.209 \pm 0.002$ |
| Other-Black | Income | $0.724 \pm 0.004$ | $0.628 \pm 0.006$ | $0.216 \pm 0.007$ | $0.143 \pm 0.004$ |
| Other-Mixed | Income | $0.707 \pm 0.005$ | $0.593 \pm 0.005$ | $0.169 \pm 0.006$ | $0.117 \pm 0.004$ |
| Asian-Black | Income | $0.664 \pm 0.008$ | $0.587 \pm 0.004$ | $0.142 \pm 0.005$ | $0.111 \pm 0.004$ |
| Asian-Mixed | Income | $0.644 \pm 0.005$ | $0.607 \pm 0.006$ | $0.159 \pm 0.008$ | $0.128 \pm 0.006$ |
| White-Mixed | Income | $0.613 \pm 0.005$ | $0.546 \pm 0.005$ | $0.082 \pm 0.004$ | $0.058 \pm 0.002$ |
| White-Black | Income | $0.613 \pm 0.005$ | $0.57 \pm 0.007$ | $0.113 \pm 0.008$ | $0.08 \pm 0.006$ |
| Black-Mixed | Income | $0.603 \pm 0.006$ | $0.523 \pm 0.007$ | $0.055 \pm 0.007$ | $0.023 \pm 0.004$ |
| Asian-Black | TravelTime | $0.677 \pm 0.052$ | $0.502 \pm 0.011$ | $0.021 \pm 0.009$ | $0.01 \pm 0.003$ |
| Asian-Other | TravelTime | $0.653 \pm 0.024$ | $0.528 \pm 0.006$ | $0.053 \pm 0.011$ | $0.027 \pm 0.004$ |
| Asian-Mixed | TravelTime | $0.647 \pm 0.013$ | $0.557 \pm 0.003$ | $0.096 \pm 0.004$ | $0.045 \pm 0.002$ |
| White-Other | TravelTime | $0.636 \pm 0.02$ | $0.568 \pm 0.007$ | $0.107 \pm 0.01$ | $0.06 \pm 0.005$ |
| Other-Mixed | TravelTime | $0.618 \pm 0.017$ | $0.546 \pm 0.011$ | $0.079 \pm 0.012$ | $0.043 \pm 0.006$ |
| Other-Black | TravelTime | $0.615 \pm 0.021$ | $0.526 \pm 0.011$ | $0.049 \pm 0.014$ | $0.026 \pm 0.006$ |
| White-Black | TravelTime | $0.599 \pm 0.006$ | $0.569 \pm 0.004$ | $0.12 \pm 0.006$ | $0.057 \pm 0.003$ |
| Black-Mixed | TravelTime | $0.588 \pm 0.012$ | $0.557 \pm 0.012$ | $0.098 \pm 0.015$ | $0.0557 \pm 0.001$ |
| White-Mixed | TravelTime | $0.557 \pm 0.008$ | $0.497 \pm 0.006$ | $0.016 \pm 0.004$ | $0.006 \pm 0.002$ |
| Other-Black | Employment | $0.744 \pm 0.008$ | $0.524 \pm 0.005$ | $0.036 \pm 0.005$ | $0.036 \pm 0.004$ |
| Asian-Other | Employment | $0.711 \pm 0.011$ | $0.557 \pm 0.003$ | $0.066 \pm 0.004$ | $0.066 \pm 0.003$ |
| White-Other | Employment | $0.695 \pm 0.007$ | $0.524 \pm 0.003$ | $0.019 \pm 0.005$ | $0.019 \pm 0.002$ |
| Other-Mixed | Employment | $0.683 \pm 0.022$ | $0.557 \pm 0.008$ | $0.083 \pm 0.005$ | $0.083 \pm 0.003$ |
| Black-Mixed | Employment | $0.678 \pm 0.028$ | $0.534 \pm 0.007$ | $0.049 \pm 0.007$ | $0.048 \pm 0.004$ |
| Asian-Mixed | Employment | $0.671 \pm 0.019$ | $0.61 \pm 0.006$ | $0.0144 \pm 0.006$ | $0.145 \pm 0.004$ |
| Asian-Black | Employment | $0.655 \pm 0.021$ | $0.587 \pm 0.004$ | $0.106 \pm 0.006$ | $0.106 \pm 0.004$ |
| White-Mixed | Employment | $0.651 \pm 0.009$ | $0.581 \pm 0.006$ | $0.095 \pm 0.004$ | $0.095 \pm 0.003$ |
| White-Black | Employment | $0.619 \pm 0.011$ | $0.544 \pm 0.004$ | $0.049 \pm 0.003$ | $0.049 \pm 0.002$ |
| Asian-Mixed | Mobility | $0.753 \pm 0.02$ | $0.511 \pm 0.014$ | $0.04 \pm 0.012$ | $0.014 \pm 0.006$ |
| Other-Mixed | Mobility | $0.748 \pm 0.02$ | $0.573 \pm 0.015$ | $0.113 \pm 0.017$ | $0.062 \pm 0.009$ |
| Asian-Other | Mobility | $0.714 \pm 0.011$ | $0.565 \pm 0.01$ | $0.114 \pm 0.011$ | $0.054 \pm 0.005$ |
| Asian-Black | Mobility | $0.672 \pm 0.012$ | $0.503 \pm 0.014$ | $0.032 \pm 0.011$ | $0.012 \pm 0.004$ |
| Other-Black | Mobility | $0.66 \pm 0.012$ | $0.526 \pm 0.009$ | $0.044 \pm 0.009$ | $0.02 \pm 0.004$ |
| White-Mixed | Mobility | $0.655 \pm 0.007$ | $0.568 \pm 0.005$ | $0.105 \pm 0.007$ | $0.044 \pm 0.003$ |
| White-Other | Mobility | $0.626 \pm 0.017$ | $0.555 \pm 0.009$ | $0.091 \pm 0.01$ | $0.046 \pm 0.005$ |
| White-Black | Mobility | $0.611 \pm 0.009$ | $0.518 \pm 0.008$ | $0.043 \pm 0.008$ | $0.017 \pm 0.004$ |
| Black-Mixed | Mobility | $0.602 \pm 0.035$ | $0.503 \pm 0.016$ | $0.031 \pm 0.013$ | $0.012 \pm 0.006$ |

## F   LIME as an Alternative to Shapley Values

The definition of ET (Def. 3.2) is parametric in the explanation function. We used Shapley values for their theoretical advantages (see Appendix A). Another widely used feature attribution technique is LIME (Local Interpretable Model-Agnostic Explanations). The intuition behind LIME is to create a local linear model that approximates the behavior of the original model in a small neighbourhood of the instance to explain (Ribeiro et al., 2016b;a), whose mathematical intuition is very similar to the Taylor/Maclaurin series. This section discusses the differences in our approach when adopting LIME instead of the SHAP implementation of Shapley values. First of all, LIME has certain drawbacks:

- **Computationally Expensive:** Its current implementation is more computationally expensive than current SHAP implementations such as TreeSHAP (Lundberg et al., 2020), Data SHAP (Kwon et al., 2021; Ghorbani & Zou, 2019), or Local and Connected SHAP (Chen et al., 2019). This problem is exacerbated when producing explanations for multiple instances (as in our case). In fact, LIME requires sampling data and fitting a linear model, which is a computationally more expensive approach than the aforementioned model-specific approaches to SHAP. A comparison of the execution time is reported in the next sub-section.

- **Local Neighborhood:** The randomness in the calculation of local neighbourhoods can lead to instability of the LIME explanations. Works including Slack et al. (2020); Alvarez-Melis & Jaakkola (2018); Adebayo et al. (2018) highlight that several types of feature attributions explanations, including LIME, can vary greatly.

- **Dimensionality:** LIME requires, as a hyperparameter, the number of features to use for the local linear model. For our method, all the features in the explanation distribution should be used. However, linear models suffer from the curse of dimensionality. In our experiments, this is not apparent, since our synthetic and real datasets are low-dimensional.

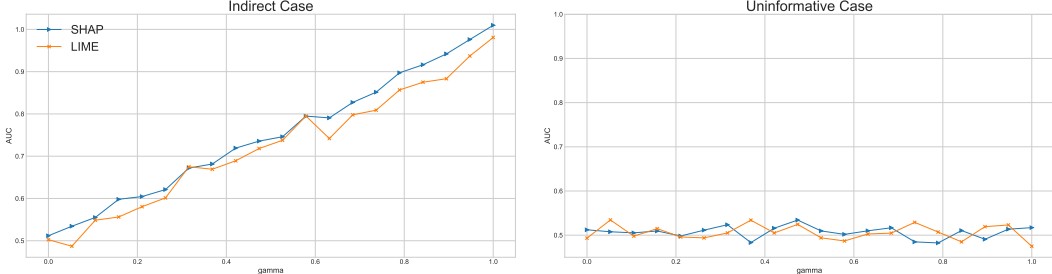

Figure 10: AUC of the ET inspect using SHAP vs using LIME.

Figure 10 compares the AUC of the ET inspector using SHAP and LIME as explanation functions over the synthetic dataset of Section 5.1. In both scenarios (indirect case and uninformative case), the two approaches have similar results. In both cases, however, the stability of using SHAP is better than using LIME.

### F.1   Runtime

We conduct an analysis of the runtimes of generating the explanation distributions using TreeShap vs LIME. We adopt `shap` version 0.41.0 and `lime` version 0.2.0.1 as software packages. In order to define the local neighborhood for both methods in this example, we used all the data provided as background data. The model $f_\theta$ is set to `xgboost`. As data we produce a randon generated data matrix, of varying dimensions. When varying the number of samples, we use 5 features, and when varying the number of features, we use 1000 samples. Figure 11 shows the elapsed time for generating explanation distributions with varying numbers of samples and columns.

The runtime required to generate explanation distributions using LIME is considerably greater than using SHAP. The difference becomes more pronounced as the number of samples and features increases.

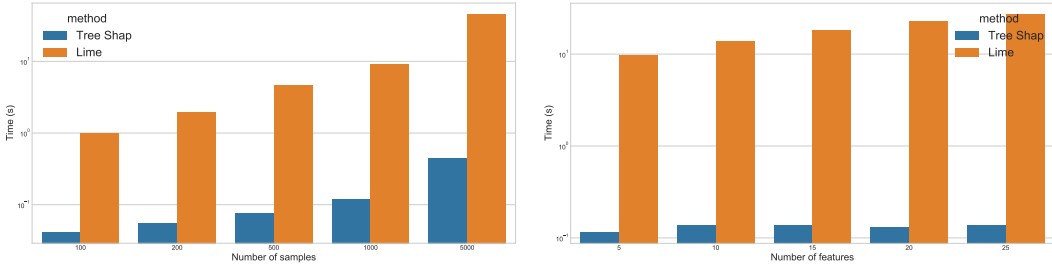

Figure 11: Elapsed time for generating explanation distributions using SHAP and LIME with different numbers of samples (left) and features (right) on synthetically generated datasets. Note that the y-scale is logarithmic.

