# OpenReview forum: "Beyond Demographic Parity: Redefining Equal Treatment"
_ICLR.cc/2024/Conference — Submitted to ICLR 2024_

### Official Review · Reviewer_pKVE · 2023-10-30

**Soundness:** 2 fair
**Presentation:** 2 fair
**Contribution:** 2 fair
**Rating:** 3
**Confidence:** 3

**Summary:**

The paper proposes a new notion of equal treatment (ET) which requires the model’s explanations to be independent from the sensitive attribute, as opposed to the demographic parity (DP) that require the independence of model prediction and sensitive attribute. Given the proposed notion, the paper first explores the relation between ET and DP, and then proposes a method to inspect ET via statistical independence test.  Such an inspector may further help interpret the sources of unequal treatment.

**Strengths:**

1. Inspecting whether a model violates fairness and explaining the sources that cause unfairness is an important and interesting problem.
2. The paper proposed a new notion of fairness based on explanation distribution, which is novel to the best of my knowledge.
3. The paper validates the proposed inspector on both synthetic and real data.

**Weaknesses:**

1. The main concern I had was the novelty of the paper, which I think is not sufficient. Specifically, using model attribution methods such as Shapley values to interpret model unfairness has been explored in prior works; the idea of using the two-sample test to examine the independence of two sets of variables has also been studied. While the settings are not the same, the techniques are somehow similar.
2. Because the notion of equal treatment is strictly stronger than the demographic fairness notion, it can be much more challenging to attain ET in practice than DP. Moreover, the trade-off between fairness and accuracy may make ET less suitable for real applications. While the paper has compared the two notions, it is still not convincing why equal treatment is a superior notion. It is helpful if authors can provide more justification with a real example.
3. While the settings with non-linear models and non-i.i.d. data are considered in experiments, most theoretical results and illustrating examples are limited to linear models and i.i.d. data. Moreover, the synthetic data used in the evaluation is also very simple: logistic model with Gaussian distributed data.
4. The paper is not using the ICLR template.

**Questions:**

1. It seems that the sources of unequal treatment can only be explained for linear models (as illustrated in Example 4.4 and Figure 3). How can the method be generalized to non-linear cases?
2. Since ET can be much more difficult to achieve, can you provide a real example to illustrate why equal treatment is a more appropriate notion than demographic parity?

---

> ### Author Response · Authors · 2023-11-17
>
> Dear reviewer, many thanks for the comments.
> Please consider the main points of our rebuttal.
> We address specific points below:
>
>
> > I.1. The main concern I had was the novelty of the paper, which I think is not sufficient. Specifically, using model attribution methods such as Shapley values to interpret model unfairness has been explored in prior works; the idea of using the two-sample test to examine the independence of two sets of variables has also been studied. While the settings are not the same, the techniques are somehow similar.
>
> A.1. The intersection of fairness and explainable AI has been an active topic in recent years. We compare our approach with related work in Appendix B3. We would thank the reviewer for any  additional paper to be considered.
>
>
> > The paper is not using the ICLR template.
>
> We use the following in the latex preamble: \usepackage{iclr2024_conference}
>
>
> >I.2. It seems that the sources of unequal treatment can only be explained for linear models (as illustrated in Example 4.4 and Figure 3). How can the method be generalized to non-linear cases?
>
> A.2. The approach is not restricted to linear models only. For instance, in Appendix E.3 non-linear models are experimented with. If the ET Inspector is linear, then the explanations are readily available, but, in general, the explanation of a non-linear ET inspector is proposed in the process from Figure 1.
>
>
> > I.3. Since ET can be much more difficult to achieve, can you provide a real example to illustrate why equal treatment is a more appropriate notion than demographic parity?
>
>
>  A.3 The main philosophical critique to the liberalism notion of equal of treatment as blindness, is that it is rarely achieved. Our paper confirms this classical intuition both, theoretically and experimentally.
>
> In Figure 3, we provide cases where Demographic Parity is achieved but Equal Treatment is not, these are cases where demographic parity may flag wrongly a fair ML model. This figure is further extended in table 5 where we compare against distinct methods to measure demographic parity.

---

### Official Review · Reviewer_9JSK · 2023-10-31

**Soundness:** 3 good
**Presentation:** 3 good
**Contribution:** 3 good
**Rating:** 8
**Confidence:** 3

**Summary:**

This paper proposed a new fairness definition motivated by the pursuit of equal treatment. The authors first showed that it is insufficient to use statistical measures of equal outcome, e.g. demographic parity, to evaluate equal treatment. They then defined Equal Treatment (ET) as requiring indistinguishable explanation distribution for the non-protected features between populations with different protected features. The explanation distribution relies on an explanation function, for which Shapley value is used as the example in the paper, to quantity how non-protected features affect the trained model. Based on the new ET definition, they also designed a Classifier Two Sample Test (C2ST) to test whether a ML model provides equal treatment based on the AUC of the model. In numerical experiments, the authors demonstrated that the new ET definition is more effective at inspecting treatment equality in a model, and their method could provide explanation for the underlying causes of treatment inequality.

**Strengths:**

The proposed Equal Treatment is a novel method that combines fairness and explainability. These two goals are both important components in the broad domain of ethical machine learning, and they are typically studied separately. The Equal Treatment Inspector workflow from this paper examines both issues and can answer the useful question of what causes unfairness.

The paper is well-written and follows a well-thought-out flow. The examples provided throughout the paper are helpful for understanding the concept. The related works (majority in appendix) are thoroughly reviewed to help position the paper in literature.

**Weaknesses:**

I disagree with some statements that the paper used to motivate the research question. For example, the abstract states “Related work in machine learning has translated the concept of equal treatment into terms of equal outcome and measured it as demographic parity (also called statistical parity)”. To my understanding, it is well-recognized in the fair ML literature that equal treatment and equal outcome are different concepts. While I agree that equal outcome is often measured with statistical measures, I think it is inaccurate to frame equal outcome as a convenient proxy of equal treatment. Instead, one simplified interpretation of equal treatment is “fairness through unawareness” or “colorblindness”. Rather than relying on the distinction between ‘equal outcome vs. equal treatment’, which can refer to much more high-level philosophical differences than what is captured in this paper, I would find it clearer to simply focus on equal treatment (new definition) vs. demographic parity.

**Questions:**

1.	In Section 4, the theoretical analysis relies on assuming exact calculations of Shapley value are available. How realistic is this assumption in practice? When we do not have access to exact Shapley values, how will the theoretical results be affected?

2.	What are other explanation functions that can be used in the framework? In the appendix, another example is given, but I wonder is there a large set of options or is designing an effective explanation function an open question itself? If there are multiple candidate explanation functions, what makes one function better than another?

---

> ### Author Response · Authors · 2023-11-17
>
> Dear reviewer, many thanks for the comment, please consider the main points of our rebuttal above and our specific points that follow now:
>
> > I.1. I disagree with some statements that the paper used to motivate the research question. For example, the abstract states “Related work in machine learning has translated the concept of equal treatment into terms of equal outcome and measured it as demographic parity (also called statistical parity)”. To my understanding, it is well-recognized in the fair ML literature that equal treatment and equal outcome are different concepts. While I agree that equal outcome is often measured with statistical measures, I think it is inaccurate to frame equal outcome as a convenient proxy of equal treatment. Instead, one simplified interpretation of equal treatment is “fairness through unawareness” or “colorblindness”. Rather than relying on the distinction between ‘equal outcome vs. equal treatment’, which can refer to much more high-level philosophical differences than what is captured in this paper, I would find it clearer to simply focus on equal treatment (new definition) vs. demographic parity.
>
> A.5.1 We agree with the reviewer. We will compare equal treatment vs demographic parity instead of equal outcomes.
>
>
> > I.2 In Section 4, the theoretical analysis relies on assuming exact calculations of Shapley value are available. How realistic is this assumption in practice? When we do not have access to exact Shapley values, how will the theoretical results be affected?
>
> I.2. The mathematical derivation of Shapley values in real cases, (non-iid data and non linear model), is not feasible. There are related works that focus on to improve the computation of Shapley values in those scenarios:
>
> [1] Sebastian Bordt, Ulrike von Luxburg: From Shapley Values to Generalized Additive Models and back. pdf. AISTATS, 2023
>
> [2] Interventional shap values and interaction values for piecewise linear regression trees, AAAI 2023
>
>
>
> > I.3. What are other explanation functions that can be used in the framework? In the appendix, another example is given, but I wonder is there a large set of options or is designing an effective explanation function an open question itself? If there are multiple candidate explanation functions, what makes one function better than another?
>
> A.3. In this work, we have focused on feature attribution explanations that are well-developed for tabular data. We have selected LIME and Shapley because they comply with the efficiency property (equation 6), and this is required so the theoretical analysis hold. It is possible that alternative XAI explanation techniques, such as logical reasoning, argumentation, or counterfactual explanations, could be useful and offer their unique advantages. This remains a further work to explore. (This issue is discussed in the “Limitations” paragraph of the Conclusions.)

---

> > ### Comment · Reviewer_9JSK · 2023-11-20
> > **Thank you for the responses**
> >
> > I thank the authors for responding to my comments.

---

### Official Review · Reviewer_ZXLv · 2023-11-08

**Soundness:** 3 good
**Presentation:** 3 good
**Contribution:** 3 good
**Rating:** 5
**Confidence:** 3

**Summary:**

The paper questions the traditional approach of equal outcome and demographic parity as a measure of fairness and proposes a new formalization for equal treatment. The authors measure equal treatment by accounting for the influence of feature values on model predictions. They formalize equal treatment by considering the distributions of explanations and comparing them between populations with different protected features. The paper proposes a classifier two-sample test based on the AUC (Area Under the Curve) of an equal treatment inspector, which compare the degree of equal treatment between different groups. The application on synthetic and real datasets show that this new equal treatment definition might actually yield higher AUCs for downstream classifiers than when using demographic parity.

**Strengths:**

- The paper is well-written and well-structured.
- It appears to be the first clear attempt to connect explanations with algorithmic fairness through the introduction of the new "equal treatment" definition. While other approaches have used explainability as a proxy for fairness, none have established such strong foundations as presented by the authors.
- The examples with simple linear models effectively illustrate potential impacts and counterexamples.
- The experiments provide compelling evidence of the potential implications of this novel "equal treatment" definition.

**Weaknesses:**

The main weaknesses I can observe are (a) practical implications of the new equal treatment definition and (b) the novelty and implication of using a classifier-two-sample test.

(a) I agree with the authors that in the case of exact demographic parity (independence), then this definition of equal treatment works (Lemma 4.2). However, my concerns arise in cases where the demographic parity is violated only by a small amount, which is the case in practice; no (useful) algorithm has a demographic parity of exactly zero, and most of the decision making algorithms usually have a small violation tolerance. Can the authors comment how equal treatment can be used on bounding demographic parity, or whether there exists any relationship there? This scenario is important for e.g., credit lending scenarios; in the U.S., the Equal Credit Opportunity Act [2] enforces no discrimination *on the outcomes* of the decision-making algorithm. From a law standpoint, one might not necessarily mind different explanations as long as the outcomes are not too dissimilar (i.e., low demographic parity).

(b) First of all, unfortunately using AUC as a test statistic for classifier-two-sample test is not novel, see [1] for example (the good thing is that AUC is a relatively well behaved statistic, so that does not change the framework). By using a C2ST in the framework, we introduce (i) a data-driven algorithm to judge the level of equal treatment in the data but also (ii) an additional notion of uncertainty in our fairness definition. For (i), in practice this means that this approach is not necessarily low-sample-size friendly (as it does not use permutations), the complexity of the classifier directly affects type I and type II error and results may vary considerably according to which classifier is chosen (which the authors have actually explored in the Appendix). For (ii), we are rejecting the null-hypothesis with a certain probability threshold, as opposed to provide a single (deterministic) number as in demographic parity. That is, we are now guaranteeing that "up to a level 1-\alpha" the algorithm is providing equal treatment. Citing again the equal credit opportunity act of 1961, such a definition of fairness would not be admissible in a credit lending scenario, which puts into question once again the practical feasibility of this new definition of equal treatment.

[1] Model-independent detection of new physics signals using interpretable SemiSupervised classifier tests, Chakravarti, Purvasha and Kuusela, Mikael and Lei, Jing and Wasserman, Larry, The Annals of Applied Statistics, 2023
[2] https://www.justice.gov/crt/equal-credit-opportunity-act-3#:~:text=prohibits%20creditors%20from%20discriminating%20against,under%20the%20Consumer%20Credit%20Protection


For completeness, I am personally unsure whether ICLR is the best venue to reach the type of audience who would be interested in this work. However, I believe this is not my judgment call to make; I assure the authors I did not take this into account when writing this review.

**Questions:**

I have included two points in the "Weaknesses" section above, so I'd be grateful if the authors would post their comments to those.

Minor points:
- The word "natural data" sounds a bit weird, as usually the machine learning community uses "real data".
- Figure 2 is a bit too small overall, increase the font size and marker size would go a long way.

---

> ### Author Response · Authors · 2023-11-17
>
> Dear reviewer, many thanks for the comments, above the main points of our rebuttal and our specific points that follow now:
>
> > I.1(a) I agree with the authors that in the case of exact demographic parity (independence), then this definition of equal treatment works (Lemma 4.2). However, my concerns arise in cases where the demographic parity is violated only by a small amount, which is the case in practice; no (useful) algorithm has a demographic parity of exactly zero, and most of the decision making algorithms usually have a small violation tolerance.  Can the authors comment how equal treatment can be used on bounding demographic parity, or whether there exists any relationship there? This scenario is important for e.g., credit lending scenarios; in the U.S., the Equal Credit Opportunity Act [2] enforces no discrimination on the outcomes of the decision-making algorithm. From a law standpoint, one might not necessarily mind different explanations as long as the outcomes are not too dissimilar (i.e., low demographic parity).
>
> A.1  Thank you for this comment. Lemma 4.2 shows that Equal Treatment implies Demographic Parity in the case of perfect independence. Apart from the trivial case where the turns out to coincide (Lemma 4.1), we have not explored in the paper bounds on DP given bounds on ET. This is something that we intend to do in future work, as it is a non-trivial property relating multivariate independence (ET) to univariate independence (DP). A promising approach will be to exploit the linearity property of Shapley values.
>
>
>
> > I.2. (b) First of all, unfortunately using AUC as a test statistic for classifier-two-sample test is not novel, see [1] for example (the good thing is that AUC is a relatively well behaved statistic, so that does not change the framework). By using a C2ST in the framework, we introduce (i) a data-driven algorithm to judge the level of equal treatment in the data but also (ii) an additional notion of uncertainty in our fairness definition. For (i), in practice this means that this approach is not necessarily low-sample-size friendly (as it does not use permutations), the complexity of the classifier directly affects type I and type II error and results may vary considerably according to which classifier is chosen (which the authors have actually explored in the Appendix). For (ii), we are rejecting the null-hypothesis with a certain probability threshold, as opposed to provide a single (deterministic) number as in demographic parity. That is, we are now guaranteeing that "up to a level 1-\alpha" the algorithm is providing equal treatment.
>
> A.2. Many thanks for pointing to relevant related work; we will integrate it on our work as it helps to clarify the strength of our submission.
>
>
> > I.3.Citing again the equal credit opportunity act of 1961, such a definition of fairness would not be admissible in a credit lending scenario, which puts into question once again the practical feasibility of this new definition of equal treatment.
>
>
> A.3. May we ask in which section of the ECOA of 1961 does the regulation state this?
>
> Note that equal opportunity, if understood by the Rawlsian "luck" egalitarianism, may fall closer to our notion than to TPR differences.
>
> In the more recent regulation discussion [3,4] Wachter advocates for demographic parity. It is important to note the EU and US law are different. And there may be other worldwide incoming regulations from other countries that might be also different.
>
> The applicability of AI regulation is an ongoing discussion with recent workshops specifically targetting this area (https://regulatableml.github.io/)
>
> In [4]""In particular, some of the tests used by the European Court of Justice and Member State courts to measure indirect discrimination match the metric of demographic parity from
> algorithmic fairness.""
>
>
>
> [1] Model-independent detection of new physics signals using interpretable SemiSupervised classifier tests, Chakravarti, Purvasha and Kuusela, Mikael and Lei, Jing and Wasserman, Larry, The Annals of Applied Statistics, 2023
> [2] https://www.justice.gov/crt/equal-credit-opportunity-act-3#:~:text=prohibits%20creditors%20from%20discriminating%20against,under%20the%20Consumer%20Credit%20Protection
>
> [3] Why fairness cannot be automated: Bridging the gap between EU non-discrimination law and AI https://www.sciencedirect.com/science/article/abs/pii/S0267364921000406
>
> [4] Bias Preservation in Machine Learning: The Legality of Fairness Metrics Under EU Non-Discrimination Law
> https://papers.ssrn.com/sol3/papers.cfm?abstract_id=3792772
>
>
>
> > I4. The word "natural data" sounds a bit weird, as usually the machine learning community uses "real data".
>
> A.4 The opposite to real data, is unreal data. While the contrary of natural data, is synthetic. We made this choice as we believe it's an improvement wrt terminology precision.

---

> > ### Comment · Reviewer_ZXLv · 2023-11-22
> > **Acknowledged Response**
> >
> > With this message I acknowledge the response of the author.
> > The response is centered around the philosophy behind the paper, to which, unfortunately, I am in no position to effectively comment on.
> > Given that (i) using AUC as test statistic is not novel and (ii) all my comments on using a CS2T and related statistical issues were not addressed, I unfortunately don't think this work is quite ready to be published, especially in a technical conference like ICLR.

---

> > > ### Author Response · Authors · 2023-11-22
> > >
> > > Dear reviewer,
> > >
> > > We appreciate your point bringing to our attention the novelty of using AUC as a C2ST test and the relevant paper from 2023. We sincerely apologize for any oversight on our part and we will promptly update the manuscript to incorporate this information.
> > >
> > > Regarding your comments, we would like to clarify that in Appendix E.1, we have presented statistical testing with the necessary parameters, and in Appendix E.5, we have included additional experiments using bootstrapping to address the raised concerns. We hope these sections sufficiently address the points you raised.

---

### Official Review · Reviewer_fCKF · 2023-11-10

**Soundness:** 2 fair
**Presentation:** 2 fair
**Contribution:** 2 fair
**Rating:** 3
**Confidence:** 4

**Summary:**

This paper highlights the issue with current fairness notions, which emphasize equal outcomes rather than equal treatment. The philosophical definition of fairness aligns more closely with the principle of equal treatment. The paper delves into the theoretical relationship between equal treatment and equal outcomes and introduces a methodology for assessing equal treatment.

**Strengths:**

High-level Idea is simple and intuitive.

**Weaknesses:**

- [Major] I remain unconvinced that 'equal treatment' is a superior notion of fairness. The paper advocates for the use of Shapley values to distribute explanations when defining equal treatment. However, the rationale for this preference is unclear to me. A notable limitation of this fairness notion is its potential indirect correlation with protected attributes like Z. For example, height is often closely associated with gender. Therefore, a model's Shapley values may not depend on the protected attribute Z and might predominantly base predictions on height equally across different gender groups, which superficially appears gender-neutral and meets the paper's fairness criteria, yet it may still result in substantial unfairness. I welcome corrections if my understanding of Shapley values is inaccurate.
- [Major] The paper omits a critical discussion on related work. There appears to be a study, specifically on individual fairness [1], that resonates with the motivations of this paper. Individual fairness emphasizes that individuals with similar backgrounds (e.g., salary, job status) should receive similar treatment. However, this paper does not draw any comparisons with its own concept of fairness to that of individual fairness.
- [Medium] The motivation presented within the paper is somewhat unclear, and it is concerning that significant discussions related to the work are relegated to the appendix. This decision diminishes the visibility and importance of such discussions.
- [Medium] I cannot agree with the authors that equal opportunity could lead to reverse discrimination and overcorrection. As far as I know, equal opportunity is proposed to address these limitations you mentioned which suffered by demographic parity. Can you cite the corresponding works that draw this conclusion?
- [Medium] Although Shapley values are central to defining 'equal treatment', they are introduced late in the appendix. It is my suggestion that the authors reconsider the organization of the paper, as many pivotal elements seem to be understated by their placement in the appendix.

Reference:

- Cynthia Dwork, Moritz Hardt, Toniann Pitassi, Omer Reingold, and Richard S. Zemel. Fairness through awareness. In ITCS, pp. 214–226. ACM, 2012.

**Questions:**

See the weakness above.

---

> ### Author Response · Authors · 2023-11-17
>
> Dear reviewer, many thanks for the comments, see the main and specific rebuttal:
>
>
> >I1.1 [Major] I remain unconvinced that 'equal treatment' is a superior notion of fairness.
>
> A.1.1.In the main rebuttal, we have argued why Equal Treatment is a superior notion to Demographic Parity. In Appendix B1, we have provided an illustrative example regarding blind reviews of papers
>
>
> >I1.2  A notable limitation of this fairness notion is its potential indirect correlation with protected attributes like Z....
>
> A 1.2.In example 4.4 we have studied the indirect correlation with protected attributes, which is derived from the Shapley value theoretical propertie (Appendix A). Here, we adapt the example to the case that the reviewer proposes. We will predict a fictitious variable “weight” based on “age” and “place of birth” (both independent of “weight”) and “height” which is related to “weight”.
>
> Let $X = X_1,X_2,X_3$ be independent features, and $X_1, X_2 \perp Z$, and $X_3 \not \perp Z$. Where $X_1$ and $X_2$ are “place of birth” and “age” respectively, and $X_3$ is “height” which correlates with gender $Z$ and “weight” $Y$.
>
> We now have a linear model $f_\beta(x_1, x_2, x_3) = \beta_0 + \beta_1 \cdot x_1 + \beta_2 \cdot x_2 + \beta_3 \cdot x_3$ with $\beta_1, \beta_2, \beta_3 \neq 0$.
>
> We can calculate the Shapley values by $S(f_\beta, x)_i = \beta_i \cdot x_i$. (Aas,2021)
>
> Now we measure Equal Treatment by $g_\psi(s) = \psi_0 + \psi_1 \cdot s_1 + \psi_2 \cdot s_2+\psi_3 \cdot s_3$, which can be written in terms of the $x$'s as: $g_\psi(x) = \psi_0 + \psi_1 \cdot \beta_1 \cdot x_1 + \psi_2 \cdot \beta_2 \cdot x_2+\psi_3 \cdot \beta_3 \cdot x_3$.
> By OLS estimation properties, we have $\psi_1 \approx cov(\beta_1 \cdot X_1, Z)/var(\beta_1 \cdot X_1) = cov(X_1, Z)/(\beta_1 \cdot var(X_1))  = 0$ and analogously $\psi_2 \approx 0$. Finally, $\psi_3 \approx cov(X_3, Z)/(\beta_3 \cdot var(X_3)) \neq 0$.
>
> The coefficients of $g_\psi$ provide information about which feature contributes to the dependence between the explanation $S(f_\beta, X)$ and the protected feature $Z$.
>
> Using the example the reviewer proposed $\psi_3$ shows how related “height” is to the protected attribute in contributing to the prediction, thus violating Equal Treatment.
>
> Many thanks to the author for the question, which helps clarify the paper's contributions. Does the reviewer find this clarification helpful?
>
> > I.2[Major] The paper omits a critical discussion on related work. There appears to be a study, specifically on individual fairness [1], that resonates with the motivations of this paper.
>
> A.2 Individual fairness can not often be applied and is therefore not a popular metric [2]– unlike group fairness metrics. For example, when a company hires for one position, it obviously has to discriminate even if two individuals are identical wrt non-protected characteristics. Talking about unfainress is not  plausible in this situation. Only if the company shows a systematic, statistically significant pattern of discriminating a group of people, one can talk about unfairness. Some Individual fairness metrics may have the challenge of achieving a precise mathematical definition of “similarity”. See [1] for a summary of critics on fairness metrics.
>
> [1] Salvatore Ruggieri, José M. Álvarez, Andrea Pugnana, Laura State, Franco Turini:
> Can We Trust Fair-AI? AAAI 2023
>
> [2] Fleisher, W. What's fair about individual fairness?. AIES 2021
>
> > I.3[Medium] The motivation presented within the paper is somewhat unclear, and it is concerning that significant discussions related to the work are relegated to the appendix. This decision diminishes the visibility and importance of such discussions.
>
> A.3. We will adapt the introduction based on the feedback of the reviewers, but, due to space constraints, we decided to privilege the main contributions in the main text, and the detailed discussions in the appendix.
>
>
> > I.4 [Medium] I cannot agree with the authors that equal opportunity could lead to reverse discrimination and overcorrection.
>
> A.3 See first answer for clarification (A.1).
>
> From a technical perspective, Equal Opportunity requires labeled data which in practice, which after deployment is hard to obtain. Equal Opportuntiy forces to meet some quotas (True Positive Rate), this introduces a second order discrimination. Equal Treatment has no quotas, it is fine to “hire” the best, while the decision is taken independently from the protected attribute.
>
> From a philosophical perspective much is discussed on the literature, the commentary at https://link.springer.com/content/pdf/10.1057/palgrave.cpt.9300060.pdf about the book [1] provides some hints.
>
>
>
> [1] Matt Cavanagh, Against Equality of Opportunity. Clarendon Press, 2002.

---

> > ### Comment · Reviewer_fCKF · 2023-11-22
> > **Thanks for your response**
> >
> > Thanks for your detailed responses.
> >
> > Re A1.2, thanks for the clarification. While this example seems to assume the features are independent but still helps.
> >
> > However, I am still not convinced that the proposed notion is better.
> >
> > Meanwhile, while I agree that the proposed fairness notion is better than the demographic parity from the aspect of philosophy, as I mentioned, there are other metrics that are more reasonable from the aspect of philosophy. Establishing some connection between the proposed fairness notion with them (not just DP) would be very helpful.
> >
> > Moreover, I cannot see how will the paper integrate the related work discussion here into the final version.
> >
> > To summarize, I re-evaluate the idea and agree that the proposed equal treatment is interesting. However, there is still some work to be done to make it ready for being published. Therefore, I will keep my original score.

---

> > > ### Author Response · Authors · 2023-11-22
> > >
> > > Dear reviewer,
> > >
> > > Thank you very much for your consideration.
> > >
> > > You have mentioned individual fairness, which has severe shortcomings for the above reasons, and we agree that they must be included in the paper.
> > >
> > > Which specific metrics does the reviewer think would be more reasonable from a philosophical point of view?
> > >
> > > best
> > > The authors

---

> > > > ### Comment · Reviewer_fCKF · 2023-11-22
> > > > **To Authors**
> > > >
> > > > The causal fairness, and individual fairness are also very reasonable fairness notions. However, as the authors mentioned, there are some limitations of causal fairness and individual fairness. Therefore, it would be very helpful if the authors could establish some connection between equal treatment and these existing "reasonable" fairness notions just like what the authors did for demographic parity, and claim that the proposed fairness notion also enjoys some empirical advantages (e.g., convenient, etc). This more thorough comparison would be very helpful.

---

> ### Author Response · Authors · 2023-11-22
>
> Dear reviewer, in the following, we relate further metrics to philosophical requirements, as you have asked for. Please let us know if further questions or doubts remain.
>
> __Individual fairness__ is specified according to the original definition by requiring that
> $d (f(x),f(x')) \leq L \cdot d(x,x')$ where L > 0 is a Lipschitz constant and
> d(·, ·) is a distance function.
>
> From a philosophical perspective, individual fairness is closest to the liberal point of view. However, it fails the requirements of liberalist arguments. Liberalism argues for meritocracy, i.e. disparate treatment is okay if it is based on varying efforts or preferences of individuals but not fair if it is based on characteristics that individuals did not choose.
> E.g. it's fair to hire someone because of better grades, but it is unfair if these grades depend on ethnicity. These varying treatments of attributes is not captured by the definition of Individual Fairness above - but it is captured by our notion of Equal Treatment.
>
> Also, from the technical perspective, the philosophical requirements are not matched: (i) a definition of distance including continuous and discrete attributes leaves a lot of leeways and leaves decisions to the choice of hyperparameters. (ii) Individual fairness is not a statistical measure and will fail to account for situations, such as hiring decisions, where one needs to decide even if the differences between individuals are minimal (also our rebuttal answer A.2)
>
>
>
> __Fairness through un-awareness__
> An algorithm is fair according to this definition of fairness if all the protected attributes $Z$ are not explicitly used in decision-making.
> Any learned function $f: X \rightarrow Y$ that excludes $Z$ satisfies this. Fairness by unawareness has the shortcoming that elements of $X$ can depend on $Z$. Such features are called proxy features. They can be used to train $f$ to make discriminatory decisions.
>
> __Counterfactual Fairness__
> Another metric iscounterfactual fairness[1]. In the literature, it has been observed that this definition is “basically demographic parity”[2]. Thus, counterfactual fairness exhibits the same drawbacks that we have elaborated about Demographic Parity
>
>
> __Causal Fairness__
> More commonly, causal fairness refers to metrics that require causal structure to be known for computing fairness or its violation. For example, Definition 1 ( taken from [3]) includes interventions to define fairness. Even if this definition may mirror what the liberal school of thought asks for. It remains a huge challenge to model or learn causal structures - and most often they remain unavailable.
>
>
>
> __To conclude__, our notion of equal treatment is a practical and useful measure for judging the amount of unfairness, following the liberal school of thought in philosophy. Dealing with philosophical and practical requirements is indeed a superior fairness metric. We acknowledge that we should have elaborated on these metrics more broadly. However, as the reviewer may see from our rebuttal, we have concise arguments as to the superiority of our metric and are happy to include them in the paper - most likely in the appendix due to length restrictions.
>
> [1] Counterfactual Fairness.Matt J. Kusner, Joshua R. Loftus, Chris Russell, Ricardo Silva: NIPS 2017
>
> [2] Counterfactual Fairness Is Basically Demographic Parity. Lucas Rosenblatt, R. Teal Witter. AAAI 2023
>
> [3] Causal feature selection for algorithmic fairness. Galhotra, S., Shanmugam, K., Sattigeri, P., & Varshney, K. R. In Proceedings of the 2022 International Conference on Management of Data.

---

### Official Review · Reviewer_1Jx6 · 2023-11-12

**Soundness:** 2 fair
**Presentation:** 2 fair
**Contribution:** 3 good
**Rating:** 6
**Confidence:** 3

**Summary:**

The authors propose an Equal Treatment Inspector that identifies features responsible for the equal treatment fairness violation.
The authors perform experiments using LIME and Shapley explanation methods and use xgboost for the models and logistic regression for the inspectors.

**Strengths:**

The authors identify an interesting problem in fair predictive decision-making.
They propose a feasible solution and perform various experiments.
In addition, authors operationalize their method, which is rare.

**Weaknesses:**

Operationalized tool: ``explanationspace `` https://explanationspace.readthedocs.io/en/latest/auditTutorial.html
- I tried out the code, and while I found it impressive, several issues made the test hard.
  - When I investigated an example: https://explanationspace.readthedocs.io/en/latest/audits.html,  I realized that installing ``explanationspace`` from  https://pypi.org/project/explanationspace/#description was effective whereas the provided step in the installation doc didn't work (https://explanationspace.readthedocs.io/en/latest/installation.html)
  - The Fairness Audits: Equal Treatment example uses ``fairtools. detector import ExplanationAudit``. I couldn't find the documentation for the functions from https://pypi.org/project/FAIRtools, and the described functions directly below the example correspond to ``explanationspace.audits.ExplanationAudit``. I changed other aspects of the code(``from fairtools.detector import ExplanationAudit`` to ``from explanationspace import ExplanationAudit`` and ``detector.fit(X, y, Z="var4")`` to ``detector.fit(X, yu, Z=X["var4"])`` and .get_auc_val() to predict_proba).
- Authors should please improve the documentation in terms of ``all`` the required packages to install (requirements.txt) and the description of results in the tutorial to facilitate easy usage and adoption.

Paper structuring and related works:
- While the authors propose an interesting perspective, the paper's structuring makes it hard to appreciate their contributions. The crucial and informative information that could have made the paper stronger is relegated to the appendix.
For example, the better experiments, presentation, explanation of results, and the description of explanation functions used, among others, are in the appendix.

- In the introduction and section 2, several introduced ideas are not well connected or explained.  There are so many ideas, it's easy to miss the gist. Additionally, the paper is more oriented towards using explanation methods (SHAP and LIME) to investigate disparities in feature importance across protected groups. However, the authors provide insufficient related work in the area and problem background.  For example, there are lots of similarities between this work and other works; ``Model Explanation Disparities as a Fairness Diagnostic``: https://arxiv.org/pdf/2303.01704.pdf, ``Explanability for fair machine learning``:https://arxiv.org/pdf/2010.07389.pdf.

Methodology
- To me, some proofs and examples seem limited and don't explore corner cases. For example, I think that the statistical independence of Z from the explanation of features is a necessary but not sufficient condition for the statistical independence of the model from Z. Additionally, in example 4.3, feature X_{3} not being statistically independent of Z and the function being a linear model makes it easy to do the proof through zeroing out that features.  In most cases, the function/model might not be linear, and the relationship between features might be complex and causal graphs hard to uncover. It seems like maybe the tool being diagnostic instead of a fixture might be a better point of view.
- Given that one might not have access to test data, would it be better to apply the ET inspector as a diagnostic on the train/val data instead?

Discussion and Experiments in the main body
- It's hard to appreciate authors' experiments and results because of the following reasons;
  - The experiment setup of 3 features and one with varied dependence on Z makes it hard to appreciate the author's contributions.
  - The authors don't provide sufficient explanations or discussion of the results.
  - Authors could have compared their experimental results to other related works and shown the impact on ET inspector and explanations on fairness on the different groups (something similar to table 5 in the appendix).

Minor or okay to address later
- Having an algorithm or bulleted procedure could have improved readability.
- For novelty, authors use AUC rather than accuracy in their C2ST instead of accuracy as previously done.  This is a bit of a tradeoff, and while the scale invariance might be good, it is damaging when inspecting other cases of fairness where one might, for example care more about false positives than false negatives.
- Given the importance of understanding the features of fairness, I think it might be important to distinguish between protected and sensitive attributes.  Not all protected features are sensitive attributes. For example, gender plays a key role in admission to single-sex schools, or age plays a crucial role in admission to age-range sports or activities.
- Reliance on Z as a binary variable is restrictive, especially since there are lots of intersectionalities.
- The explanation highly relies on f_{\theta}. It might be informative to also look at features independent of the model.

**Questions:**

While the proposed method has several similarities with ``Explanability for fair machine learning`` and ``Model Explanation Disparities as a Fairness Diagnostic`` papers, operationalizing their model has positively influenced my score.
However, issues in the writeup and code documentation negatively influenced my score. Authors should please address these issues in the weakness section.

---

> ### Author Response · Authors · 2023-11-17
>
> Dear reviewer, many thanks for the comments. Please consider the main points of our rebuttal. We address specific points below:
>
> > I.1Operationalized tool:
>
> A.1 First of all, we would like to thank the reviewer for trying out the code. We really appreciate it.
> We have now made a software release `explanationspace==0.0.2, addressing the reviewer’s concerns.
>
> We would like to distinguish between the open-source software, which we will disseminate and maintain, and the repository  of the experimental results of this paper.
>
> We have now fixed the mentioned issues so the installation and tutorial of https://explanationspace.readthedocs.io/en/latest/index.html
> works accordingly.
>
>
> >I.2. While the authors propose an interesting perspective, the paper's structuring makes it hard to appreciate their contributions.
>
> A.2 Due to space constraints, we decided to privilege the main contributions in the main text, and the detalled discussions in the appendix - bade based on what we believe it is more adequate to ICLR. We are open to any restructuring suggestions.
>
>
> > I.3. The authors provide insufficient related work in the area and problem background....
>
> A.3. Both papers are distinct from ours. They do not focus on the intersection of equality notions and ML, nor on the techniques used (Classifier Two Sample Test), proposing methods that don’t have the theoretical background solidity of our approach.
>
> We will be happy to acknowledge the recent work reported in the two arxiv papers. To the best of our knowledge, the first paper is still under review, while the second, we are unaware of its publication status. However, we hope that their absence in the submission is not held against us.
>
>
> > I.4. Some proofs and examples seem limited and don't explore corner cases.....
>
> A.4 This basic example serves us to compare with the benefits of measuring bias on the input data.
> We agree that in most cases the model might not be linear. It's in this situation where the Shapley values become of use, we have studied non-lineal models in Appendix E.2 and E.3
> Often in tabular data problems, ML models are fed with many more features than the model actually use. This method helps to deal with this problem.
>
>
> > I.5 Given that one might not have access to test data, would it be better to apply the ET inspector as a diagnostic on the train/val data instead?
>
> A.5 We agree with the reviewer on this point. ET inspector can be used to detect bias (in train/val data), and to monitor bias in the absence of labeled data. We will point out this in the paper.
>
>
> > I.7Authors could have compared their experimental results to other related works and shown the impact on ET inspector and explanations on fairness on the different groups (something similar to table 5 in the appendix)
>
> A.7.  We kindly ask the reviewer which other related works our approach should be compared to.
>
>
> > I.8. For novelty, authors use AUC rather than accuracy in their C2ST instead of accuracy as previously done. This is a bit of a tradeoff, and while the scale invariance might be good, it is damaging when inspecting other cases of fairness where one might, for example, care more about false positives than false negatives.
>
> A.8. Even if we use AUC, other metrics are also possible as far as a test statistics is available (e.g., accuracy in Lopez-Paz and Oquab 2017)
>
>
>
> > I.9Reliance on Z as a binary variable is restrictive, especially since there are lots of intersectionalities.
>
> A.9 The inspector $g$ can be a multi-class classifier. In the experiment of Figure 3, there are several ethnicities. In this case we have used a 1vsAll approach, further methods can be extended even to continuous protected attributes using regression instead of classification
>
>
> > I.10 The explanation highly relies on f_{\theta}. It might be informative to also look at features independent of the model.
>
> A.10 This is done in Example 4.3 in the Theoretical section and in the “Fairness Input” analyses in the Experimental section (e.g., Figure 2).
>
>
> > I.11 While the proposed method has several similarities with Explanability for fair machine learning and Model Explanation Disparities as a Fairness Diagnostic paper, operationalising their model has positively influenced my score. However, issues in the writeup and code documentation negatively influenced my score. Authors should please address these issues in the weakness section.
>
> A.11  We have aimed to address the reviewer’s concerns in the above discussion.
>
> We do not think that not comparing to recent work that is still currently under review should be held against us. We will add a comparison against this work in the updated version of the manuscript.

---

> > ### Comment · Reviewer_1Jx6 · 2023-11-19
> >
> > I would like to thank the authors for addressing the questions I had

---

### Author Response · Authors · 2023-11-17
**Main Rebuttal**

We thank all reviewers for their valuable and insightful comments.
We want to respond to a major concern several reviewers shared:

Main Point. Is “equal treatment” a superior notion of fairness?

Answer: To define a quality metric, one must collect domain requirements and formalise the metric such that its mathematical properties match the requirements. For the metrics of fairness, the domain requirements are given by philosophical foundations (section 2.2). For the egalitarian school of thought, we find that Hardt et al. 2016 propose a metric for equal opportunity. Still, it has the shortcoming in that it requires labelled data, for calculating the true positives and the false negatives  - which are hardly ever available in practice - thus making equal opportunity often unusable.

For the equal treatment of individuals with respect to a protected attribute, demographic parity violates philosophical requirements from the liberalism school of thought (cf Section 4). Thus, we find that our suggestion of equal treatment is a significant improvement from the domain requirements point of view. From the mathematical point of view, we show its desired behaviour from a theoretical perspective in section 4 and from an experimental perspective in section 5.

It is a particular strength of this submission that philosophical domain requirements are recognised and matched. This is the only way to avoid that one defines a metric that measures the wrong thing. Thus, indeed, we argue that equal treatment is a superior notion of fairness

We have also solved some software issues and improved the quality of the tutorial in the new Python package release `0.0.2`

---

### Meta-Review · Area_Chair_Vghp · 2023-12-12

**Metareview:**

The paper proposes a 're-definition' of equal treatment. This is motivated by the fact that the more common approach so far has been to use equal outcomes as a criterion. The authors (correctly) argue that in many cases the outcomes are not available. The notion of demographic parity seems a bit of a straw-man, even though indeed some papers use it. The papers makes heavy references to philosophy (and also in the rebuttals). My concern is that this does not fit well within the technical expertise of the reviewers for ICLR (or indeed the meta-reviewer). Although the reviews have expressed different opinions, overall I think there has been a fair discussion and in particular, I share Reviewer fCKF's concerns about lack of comparison to other notions of fairness.

**Justification For Why Not Higher Score:**

Mixed reviews and lack of enthusiasm about reviewers.

**Justification For Why Not Lower Score:**

Seems borderline.

---

### Decision · Program_Chairs · 2024-01-16

Reject